# Privacy-Preserving Surveillance as an Edge Service Based on Lightweight Video Protection Schemes Using Face De-Identification and Window Masking

**Alem Fitwi** [1], **Yu Chen** [1,*], **Sencun Zhu** [2], **Erik Blasch** [3] **and Genshe Chen** [4]

1 Department of Electrical and Computer Engineering, Binghamton University, Binghamton, NY 13902, USA; afitwi1@binghamton.edu
2 Department of Computer Science and Engineering, Penn State University, University Park, PA 16802, USA; sxz16@psu.edu
3 The U.S. Air Force Research Laboratory, Arlington, VA 22203, USA; erik.blasch@gmail.com
4 Intelligent Fusion Tech, Inc., Germantown, MD 20876, USA; gchen@intfusiontech.com
* Correspondence: ychen@binghamton.edu

**Abstract:** With a myriad of edge cameras deployed in urban and suburban areas, many people are seriously concerned about the constant invasion of their privacy. There is a mounting pressure from the public to make the cameras privacy-conscious. This paper proposes a Privacy-preserving Surveillance as an Edge service (PriSE) method with a hybrid architecture comprising a lightweight foreground object scanner and a video protection scheme that operates on edge cameras and fog/cloud-based models to detect privacy attributes like windows, faces, and perpetrators. The Reversible Chaotic Masking (ReCAM) scheme is designed to ensure an end-to-end privacy while the simplified foreground-object detector helps reduce resource consumption by discarding frames containing only background-objects. A robust window-object detector was developed to prevent peeping via windows; whereas human faces are detected by using a multi-tasked cascaded convolutional neural network (MTCNN) to ensure de-identification. The extensive experimental studies and comparative analysis show that the PriSE scheme (i) can efficiently detect foreground objects, and scramble those frames that contain foreground objects at the edge cameras, and (ii) detect and denature window and face objects, and identify perpetrators at a fog/cloud server to prevent unauthorized viewing via windows, to ensure anonymity of individuals, and to deter criminal activities, respectively.

**Keywords:** edge computing; face detector; foreground object detector; privacy-preserving; reversible scrambling; window-object detector

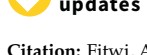

## 1. Introduction

With the widespread use of closed circuit television (CCTV) cameras, privacy has become an emerging issue [1,2]. Some people have positive views of the surveillance system because of its potential to deter and reduce crimes on top of providing footage of crime scenes or criminal activities as evidence against criminals in courts of law [3]. However, many people are concerned and/or suspicious about the daily invasion of their privacy by the pervasive deployment of cameras in public places like streets, and marketplaces. Many people want to maintain their right to determine what information about them others can collect and see. Hence, people want the surveillance system to have the ability to prevent the spill of their information to the public cyberspace—where there are about 4.57 billion users at present [4]—owning to interception attacks or abuses of the system. Generally, there are two lines of antithetical arguments in relation to the practice of mass-surveillance. On one hand, governments and pro mass-surveillance people claim that mass-surveillance engenders no harm to good people and only bad people have reasons to conceal things. They often mention the quote "*If you have got nothing to hide, you have got nothing to worry about*" to substantiate their argument. On the other hand, many argue

against that it is never all about hiding something; it is all about that being "none of other people's business". Furthermore, they quote "*I don't have anything to hide, but I don't have anything I feel like showing to you, either*" [5,6] to corroborate their position. However, the bottom line is that privacy of individuals is breached on daily basis and a multitude of people want to see privacy-conserving surveillance practices realized.

Furthermore, most of the preexisting mechanical surveillance systems rely on security personnel to observe any illicit acts, and to perform analysis of the captured videos whenever deemed necessary. The surveillance capabilities of these public safety and security officers have been drastically boosted by the state-of-the-art CCTV cameras in use today. In other words, the capabilities of the deployed cameras could be potentially exploited or abused which worsens the risk of the privacy breaches of individuals. As the surveillance cameras become more powerfully equipped with better optical and digital zooms, the more likely they are to be abused to peek via windows to gather unauthorized private information on activities or behavioral patterns of individuals and to perform voyeurism unless preventive measures are enforced by design [5,7,8]. For instance, maneuverable cameras, like pan-tilt-zoom (PTZ) cameras, could be abused and directed to intrusively spy on other people in their apartments like the case where a security guard spied on the private apartment of the German Chancellor Angela Merkel using a museum's CCTV camera [2]. The second major challenge that endangers the privacy of individuals is the possible interception of the video data by adversaries while on transmission from the edge cameras to distant servers. Thirdly, the collected video data could be leaked or abused for blackmailing, stalking, or even extorting individuals by people in charge of the system [8,9].

Most of the preexisting privacy-protection mechanisms are deployed on either the fog or cloud computing architectures; and as a result, they are not convenient for ensuring end-to-end (E2E) privacy in video surveillance. The cloud computing paradigms get the video analytics performed at distant powerful cloud machines. However, transferring raw video streams to remote cloud servers increases the risk of privacy breaches and incurs unnecessary workload to the communication networks. The edge computing migrates some intelligence and processing power to the smart cameras, which allows the processing of videos and images in real-time [1]. Furthermore, the edge-computing paradigm helps enforce privacy-preserving measures at the point where the video is created [10,11]. Likewise, the edge environment supports slenderized techniques that require less computational resources. Hence, exploiting the capabilities of the edge computing paradigm, a lightweight object scanner and scrambler modules can be developed to ensure end-to-end privacy and to anonymize individuals filmed on cameras. Another case is that windows have been used as inlets for spying on individuals while at home doing private activities by means of CCTV cameras. The publicly-known high-profile incidence is the spying on the private Berlin flat of the German Chancellor Angela Merkel [2] by a CCTV camera directed through her window from a Museum. Hence, such unauthorized and illegal surveillance practices performed by abusing CCTV cameras must be prevented by design.

In this paper, we propose a Privacy-preserving Surveillance as an Edge service (PriSE) based on lightweight video protection schemes. The PriSE scheme comprises a lightweight frame scrambling method for ensuring end-to-end privacy preceded by a computationally-light foreground object detection that runs on edge cameras. A window-object detector and an MTCNN-based face detector run on a fog/cloud server to detect and denature windows and faces to prevent prowling via windows and to ensure de-identification [12], respectively. In addition, the face embeddings of individuals caught on cameras are analyzed through matching to a database of wanted criminals by using a support vector machine (SVM) based model in order to catch fugitives from justice. The major contributions of this paper are summarized as ensues:

- Thorough investigations were carried out on how to identify privacy attributes and achieve a privacy-aware surveillance service in a resource-constrained environment, along with a comparative security and performance analysis.

- A new optimal and lightweight Reversible Chaotic Masking (ReCAM) scheme is proposed for scrambling video frames/images color-channel wise to ensure end-to-end privacy. It is more suitable for video frame/image scrambling at the edge than traditional data encryption schemes and existing chaotic schemes. ReCAM is a faster scheme when safety and resilience are proven based on standard security parameters and performance metrics, respectively.
- A simplified foreground object detection algorithm is introduced to ensure that edge-cameras forward only frames that contain foreground objects to improve the processing time and utilization of bandwidth and storage.
- Deep neural network (DNN) and machine learning (ML) based models to detect window-and-face objects and to identify fugitives from justice were developed and efficiently integrated. The window-object detector is developed to detect window objects so as to prevent images taken by maneuvering the cameras.
- A number of experimental studies and analyses were carried out, including performance analysis and a comparison with existing equivalent methods. The results verify that the proposed PriSE scheme provides an end-to-end privacy protection against possible interception attacks and it is able detect and denature window and face objects to prevent unauthorized data collection about people and identification of individuals caught on cameras. Besides, fugitives whose face features have been stored on databases are effectively identified and authorized personnel are alerted right away.

The remainder of this paper is organized as follows. Section 2 presents the main findings from the related works explored in the trajectory of this work. Section 3 describes the overall architecture of the proposed PriSE scheme. Section 4 describes how the simplified foreground object works. The detailed design analysis of the computationally-thin and secure frame-scrambling mechanism is explained in Section 5. Section 6 expounds how window and face objects are detected and masked to prevent visualization and unauthorized identification ensued by the brief presentation of the key distribution management scheme in Section 7. The details of experimental setup, discussion of comparative security parameters and performance analysis and corresponding results are elucidated in Section 8. Finally, our conclusions are presented in Section 9.

## 2. Related Works

The reported work in video privacy includes eclectic concepts and methods from multiple areas of knowledge and allied fields including CCTV surveillance systems, privacy-attribute identification schemes, image scrambling methods, and object-detection technologies, which are vital for preserving privacy in surveillance systems. As a result, thorough studies were carried out on the state-of-the-art progress of these areas and the findings are presented in the following subsections.

### 2.1. Privacy in Surveillance Systems

At present, CCTV cameras are abundantly deployed in many parts of urban areas by governments, companies, and individuals with the main goal of ensuring physical security and safety. For example, an average Londoner is estimated to be caught on CCTV cameras 300 times a day while moving around the city [5,9,10,13,14]. As a result, a great deal of information about individuals is collected by the CCTV cameras without their knowledge and consent. The best solution to protect the privacy of individuals amid the increased proliferation of fixedly deployed cameras and drones is, therefore, demanding the camera companies to make their cameras privacy-conscious by design [15–18].

Based on our thorough studies and observations, the breaches of privacy in the practice of video surveillance systems could be ascribed to two factors, namely interception of videos while in transit, abuse of CCTV cameras and videos by the operators, its indiscriminate nature, and a balance problem between privacy and usability. Most existing surveillance systems are deployed based on the cloud computing architecture. As a result,

videos could be intercepted while being transported over the network exploiting some of its vulnerabilities [17,19,20], which may cause the spilling of private data into the cyber space. Secondly, people in charge of the CCTV cameras could illicitly collect private data. Additional abuses of CCTV cameras as identified by the American Civil Liberties Union (ACLU) include criminal abuse, institutional abuse, abuse for personal purposes, discriminatory targeting, and voyeurism [21]. However, none of these papers propose a universally viable solution apart from pinpointing the privacy weakness of the surveillance system.

## 2.2. Personal Privacy Attributes

The sensitive things for which individuals demand protection are known as personal privacy attributes or generally privacy-sensitive attributes. They are the information on the video frames and images captured and conveyed by the surveillance systems that can reveal a considerable amount of personal information and activities about the identity, behavioral pattern, and daily engagements of individuals [8,21,22]. Attributes like facial features reveal substantial and powerful details vital for identifying individuals. Furthermore, transmitting frames containing some highly personal activities or situations as-is could cause a lot of social embarrassment if intercepted. People also feel nervous to think that they would be filmed by cameras while visiting some places like a medical centers. In addition, people do not want to be furtively observed via openings like windows while at home. These works attempt to define and identify privacy-sensitive objects, but they do not have privacy-preserving mechanisms. For instance, a machine-learning based privacy recommendation system for image sharing in social media was proposed in [8]. However, it gives no information how the privacy of the images of social media users is protected while being sent to a cloud server for the recommendation.

In this paper, we focus on protecting sensitive information from being intercepted and spilled into the wider space while in transit on top of masking windows and faces captured by surveillance cameras to prevent visualization and to ensure anonymity. Windows have long been used as gateways through which peeping Toms or voyeurs and other sneaky people furtively prowl to look at or capture people doing personal things at their homes by means of cameras with powerful zooming capabilities [23,24]. There are no schemes robustly designed to tackle imagery-through-window misuse of CCTV cameras to date, apart from stating it as a longstanding privacy problem.

## 2.3. Advancements in Object-Detection Methods

In the eyes of privacy-sensitive objects detection and classification, object classification and localization are vital in the effort to create privacy-preserving surveillance systems. There are a number of convoluted networks like VGG-Net [25], Res-Net [26], and many others [27–29] that primarily focus on improving accuracy using deeply involved networks. However, the recent trend of research in the area of CNN-based object-detection is directed towards designing, developing and deploying compact networks [30–32] for mobile applications. However, their performance on edge devices like Raspberry PI is not yet meeting requirements. All of the publicly available lightweight Deep neural networks (DNN) like GoogLeNet, SqueezeNet, MobileNet version 1, and MobileNet version 2 can only process less than three colored 480P frames per second (fps) on a Raspberry PI 4.

There are also many popular face-detection methods with near-human labeling accuracy developed over the decades. The Haar Cascade Face Detector [33] is the first popular method is brought into being in 2001. It is a relatively faster method with a simple architecture that can detect faces of different scales. It, however, suffers from a little bit higher false positive rates (FPR) and terribly poor performance under occlusion. Besides, it detects only frontal faces. Another widely used face detection model is based on histogram oriented gradient (HOG) features and a support vector machine (SVM) [34]. It is the fastest known method on central processing unit (CPU). While it works very well for frontal and slightly non-frontal faces under small occlusion, it fails to detect smaller faces, side faces, and does not work very well under substantial occlusion. In 2015, Google developed a DNN-based

robust FaceNet [35] that achieves 99.96% accuracy rate on the facial recognition dataset Labeled Faces in the Wild (LFW). The most popular and most accurate face detection method today that attempts to addresses both face alignment and detection problems that arise from occlusion and illumination is a Multi-Tasked Cascaded Convolutional Neural Network (MTCNN) [36,37]. It is a DNN-based architecture that comprises three neural networks (Proposal Network (P-Net), Refine Network (R-Net), and Output Network (O-Net)) connected in a cascade.

Moreover, there have been some specific attempts to leverage the advancement in machine learning and object-detection technologies to localize privacy-sensitive objects [7–9,38,39]. These works give some insight about how to build privacy-aware real-time video analytics in the surveillance systems, but they offer no viable solution for preserving privacy. Our previous work [40] focuses on preventing drone-based cameras from peeking through windows. However, it employs a lightweight but simple window-object detection method based on morphological processing which has a slightly high false positive rate (FPR) of 10.67%.

*2.4. Video Frame Enciphering Schemes*

At present, there exist a number of image scrambling techniques [41–43] that could be employed to scramble video frames; however, they need to be made lighter as to fit into a resource-constrained environment, like edge cameras. Besides, they must achieve a good and optimized balance between privacy, clarity, reversibility, security, and robustness [44]. Encryption is the most secure member of the editing class of video privacy protection schemes. It protects private information and sensitive data from unauthorized accesses, and enhances the security of communication between two parties exchanging information. However, given the real-time nature of videos, the traditional symmetric key cryptographic mechanisms like Advanced Encryption Standard (AES) are not convenient for image scrambling, mainly due to low processing speed [42].

Chaotic-encryption mechanisms offer better solutions in that they have enhanced performance, high degree of sensitivity to slight changes in initial conditions, large degree of randomness, enormous key space, aperiodicity, and high security. Recently proposed chaotic image enciphering mechanisms like [42,43] are secure but they are slower to be employed at the edge of a network. Besides, the privacy protection scheme provided in [40] does not protect or anonymize sensitive attributes other than windows while in transit. The masking method employed was meant only for blurring specific region of a frame and cannot be used for full-frame denaturing for two reasons. Firstly, it works fine computationally for masking smaller regions on a frame but it is slower to be employed for full frame applications. The use of limited number of control parameters with very big numeric values affected the computational speed. Secondly, the key space is smaller, which is only 128 bits. Next, in [45], we developed a much better lightweight minor's face denaturing scheme based on the Peter De Jong Map [46] by thoroughly analyzing and redesigning some control parameters. It has an excellent security but the masking process relies on multiplication. For smaller frame regions, its performance is relatively outstanding. However, the denaturing method cannot be employed for full frame encryption to ensure end-to-end privacy because the multiplication process takes more time and it severely degrades the overall performance. These experiences served as the driving forces for us to develop a new and more efficient methods in this paper.

**3. PriSE System Architecture**

The architecture of a surveillance service plays very vital role in process of creating a privacy-conscious practice. As a consequence, we have investigated and experimented edge, fog, and cloud based architectures. In a cloud-based setting, virtually no privacy-preserving measures are enforced because the edge-cameras' main job is creating videos and forwarding them over wide area network (WAN) to the distant analytics and storage centers [47,48]. All video frames with no regard to whether they are of interest to security

personnel or not are forwarded to the cloud servers - which creates unnecessary burden over the network. In comparison to other settings, a cloud architecture creates the highest impact on a network. Besides, there is high demand for storage because video streams from many edge-cameras are sent to a cloud center. The fog architecture often refers to private campus area networks (CAN); and as a result, only internal users can have access to it. Fog architectures have a relatively better privacy than the cloud setting; it is limited, though. In the Fog case, the traffic is isolated to an enterprise network and the distance travelled is shortened; hence, the impact on bandwidth, response delay, and storage requirement are less.

The edge-based surveillance architecture is different from the cloud and fog-based ones in that no videos are streamed to anywhere. Rather the whole video analytics is intelligently performed at the edge of the network, and only alerting signals are either emailed or short message service (SMS) texts are forwarded using Global System for Mobile Communications (GSM) modems to authorized parties [49,50]. In the edge setting, privacy could be fully ensured as no video is neither transmitted nor viewed by people. Likewise, it has no impact on the bandwidth and limited storage requirement. However, edge approaches suffer from severe performance degrading. At present, DNN-based lightweight object classification and detection methods cannot process more than 4 fps at an edge device whose computational capabilities are equivalent to the most recent release of the Raspberry PI series. Besides, part of the videos are required as evidence against criminals in courts of law, but this setting does not store such videos for later use.

By combining the attractive parts of the cloud, fog, and edge computing paradigms, we propose a hybrid surveillance architecture for PriSE scheme portrayed in Figure 1 to address the glaring inefficiencies of the aforementioned architectures through optimal trade-off between fps and bandwidth/storage usage. No privacy compromise is made in the PriSE architecture. As shown on the left and right sides of Figure 1, relatively smaller number of tasks like Frame Pre-processing, Simplified Motion Detector, and Frame Scrambling are performed at the edge. The compute-intensive DNN-based privacy-sensitive object-detection processes are performed at the fog/cloud server following the frame-unscrambling process.

In a more elaborate way, the PriSE scheme performs such major tasks as foreground object detection and video frame scrambling at the network edge, and on a distant cloud/fog server it performs frame unscrambling, window and face objects detection and masking, and fugitives classification. In other words, on the cameras, a thin and simplified motion detector detects foreground objects in order to discard useless frames. Then, all object-containing frames are scrambled and sent to the video analytics center. On the cloud/fog server, an inverse scrambling process is performed followed by window and face objects detection and masking, and fugitive identification processes. An example that illustrates how the various tasks are performed is depicted in Figure 2. The major tasks of the proposed scheme are executed in order: starting at the edge camera up to the surveillance operation center (SOC). The tasks, whose detailed explanations presented in the subsequent sections, are summarized as ensues:

- *Simplified Foreground-object Detector*: It is designed to detect the presence of foreground objects or a motion based on significant changes of pixels on a current frame in comparison to a reference frame. Hence, frames that do not contain foreground objects are discarded at the edge camera for their of no interest surveillance-wise thereby saving processing time, bandwidth, and storage.
- *Frame Privacy-preserving Scheme*: It is the backbone of the entire proposed scheme. Primarily, the privacy of individuals could be affected either when the collected raw information is intercepted while in transit from edge cameras to cloud/fog servers over communication channels or when cameras are abused by people in charge of them to collect unauthorized information about individuals, like peeping via windows. This paper tries to address the first problem through the introduction of a slender novel chaotic image scrambling technique, ReCAM. Every frame that contains an object or

objects other than the background object is encrypted at the edge camera and sent over the WAN or CAN to a cloud/fog surveillance server connected to a surveillance operation or viewing center. On arrival at the server, the unscrambling process is carried out to uncover the object-containing plain frames for subsequent processing.

- *Window and Face Objects Detection and Masking*: it is introduced to stop the collection of private information and activities of individuals at their homes through the notorious practices of furtively peeping through windows and face recognition. In this paper, frames are checked if they contain windows and faces using pertinent object-detectors. If they do, the windows and faces are reversibly masked at the server before they are forwarded to the viewing station to prevent peeping and face identification, respectively.

- *Detection and Classification of Fugitives*: This has the goal of identifying fugitives, who elude police detection, when they happen to be caught on cameras and alerting authorized parties by means SMS texts or emails. This task is performed at the fog/cloud server following face detection and face embedding processes.

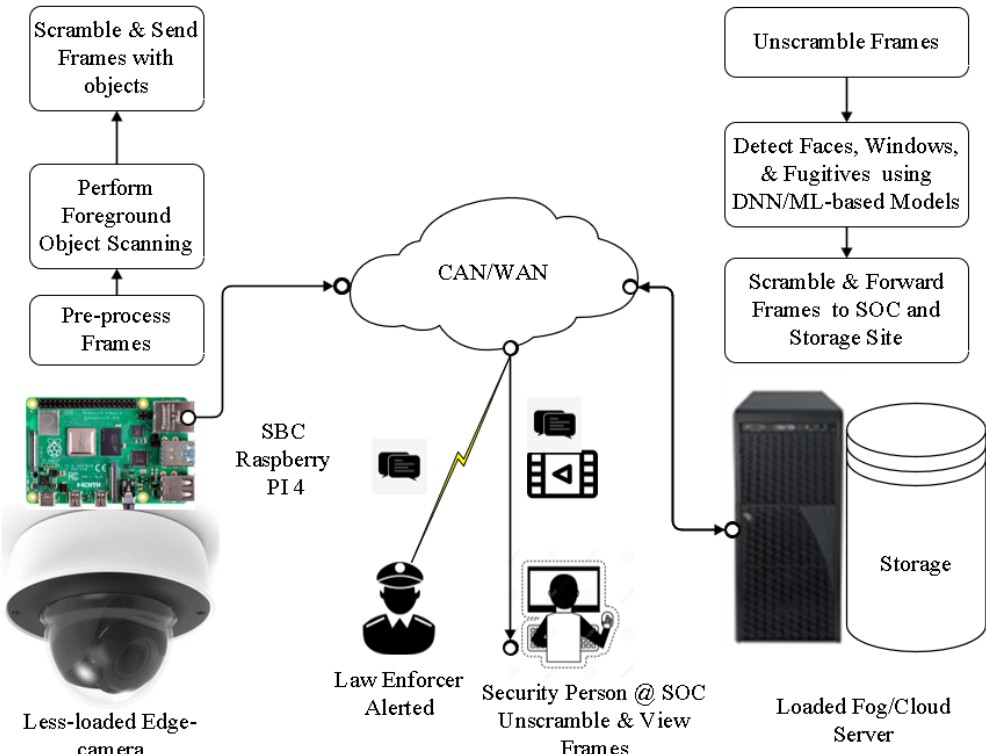

**Figure 1.** Hybrid Architecture of the PriSE system: The edge camera creates frames, performs foreground object scanning and encrypts frames. The cloud/fog server deciphers frames, detects and masks windows and faces, and performs fugitive face matching.

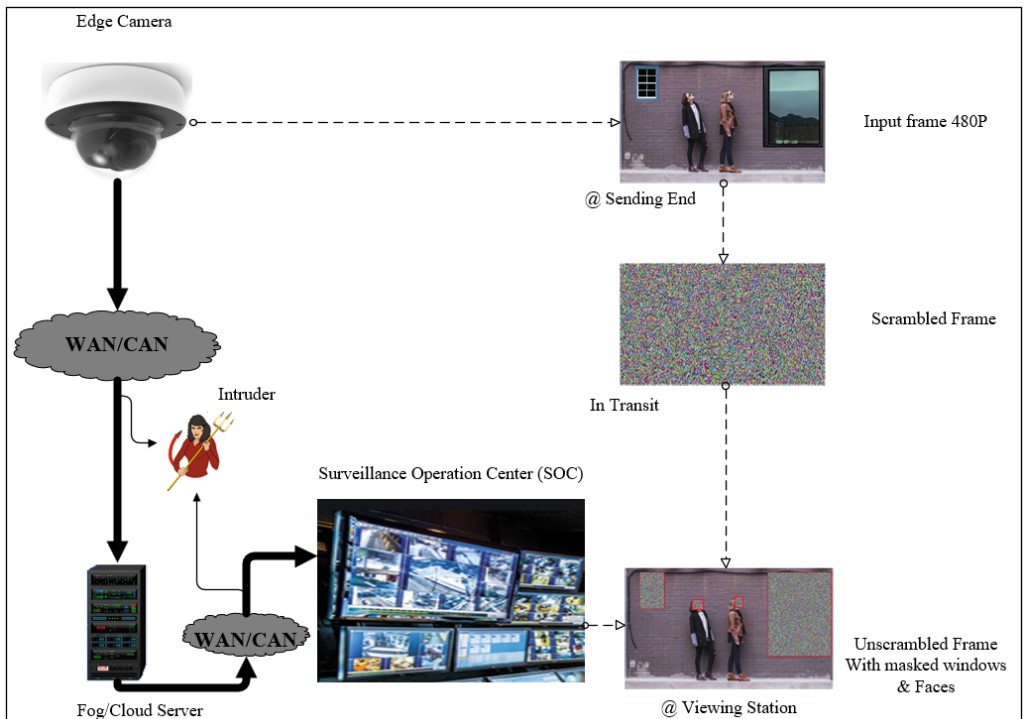

**Figure 2.** An illustration of the PriSE architecture: It comprises and edge camera that creates and encrypts frames, an intruder that tries to illegally access the video stream in transit, a cloud/fog server that deciphers frames, detects and masks windows and faces, and a viewing station where authorized people sits and observes activities caught on cameras.

## 4. Simplified Motion Detector

Video streams captured by edge cameras (wall and ceiling mounted or those perched on poles) have mostly relatively static or unchanging backgrounds over a series of video frames. Hence, it is possible to model and monitor the background for significant picture element changes. A significant change implies object motion captured by the camera. There are many sophisticated foreground and background segmentation and subtraction techniques [51,52] for motion detection. However, they are computationally expensive and infeasible for edge implementation. Hence, as illustrated in Algorithm 1, a simpler motion detector for edge implementation is proposed.

The edge cameras configured with a motion detector algorithm save latency, bandwidth and storage by discarding a number of frames with no objects or with redundant information, as illustrated in Figure 3. The first frame is used as a reference frame ($frame_{ref}$) updated every hour to account the impact of environmental factors that affect the light intensity. Then, the percentage of the difference ($frame_{diff}$) between every frame created and the reference frame is computed using a fast vector operator. The output is discriminated into black (background) and white (foreground) using a binary thresholding. Once a minimum percentage of nonzero pixels ($nz\_pixels$) in the difference frame is selected, PriSE determines whether or not to trigger the scrambling and transmission of the current frame. In our study, the minimum percentage is chosen as 0.29% experimentally. That is, the 0.29% nonzero picture elements in the difference frame is the minimum percentage that triggers the scrambling and transmission of the current frame. However, frames with nonzero pixels less than 0.29% ($nz\_pixels < 0.29\%$) are discarded. This value is made smaller to make sure that no real object is skipped undetected.

---

**Algorithm 1** @ Edge Camera Foreground Detector.

---

1: $t_0 \leftarrow$ system_time()

2: *flag* $\leftarrow 0$

3: *vid* $\leftarrow$ videoCapture()

4: *frame$_{Ref}$* $\leftarrow$ None

5: *W, H* $\leftarrow$ 640, 480 *(frame dimensions)*

6: **while** True **do**

7:     *status, frame* $\leftarrow$ vid.read()

8:     *frame* $\leftarrow$ *frame.resize((W, H))*

9:     **if** *status* **then**

10:         *frame$_{Gray}$* $\leftarrow$ to_gray(frame)

11:         *frame$_{Gray}$* $\leftarrow$ GaussianBlur(frame$_{Gray}$)

12:         $t_1 \leftarrow$ system_time()

13:         $t_2 \leftarrow t_1 - t_0$

14:         **if** *(frame$_{Ref}$ == None) or ($t_2$ == 1 h)* **then**

15:             *frame$_{Ref}$* $\leftarrow$ frame$_{Gray}$

16:             *flag* $\leftarrow 1$

17:             **if** $t_2$ *== 1 h* **then**

18:                 $t_0 \leftarrow t_1$

19:         *frame$_{Diff}$* $\leftarrow$ vectorized_xor(frame$_{gray}$, frame$_{Ref}$)

20:         *frame$_{Diff}$* $\leftarrow$ binary_threshold(frame$_{Diff}$)

21:         *pixels* $\leftarrow$ *array(frame$_{Diff}$)*

22:         *nz_pixels* $\leftarrow \frac{count\_nonzero(pixels)}{H \times W}$

23:         **if** *nz_pixels* $\geq 0.0029$ *or flag == 1* **then**

24:             **if** *flag == 1* **then**

25:                 *frame* $\leftarrow$ scramble(frame$_{Ref}$)

26:                 *flag* $\leftarrow 0$

27:             **else**

28:                 *frame* $\leftarrow$ scramble(frame)

29:         **else**

30:             *continue*

---

In Algorithm 1, $t_0$ and $t_1$ refer to the current system time and the time recorded after an hour to update the $frame_{ref}$. $t_2$ is the difference of $t_0$ and $t_1$. The to_gray() method converts a colored frame into gray-scale. The vectorized_xor() method computes the difference of the input and reference frames pairing pixel-wise in parallel. Eventually, the count_pixel() methods counts the number of non-zero pixels in the difference frame, based on which a decision is made on whether to forward or not the frame. Figure 3 portrays that only frame number 3 is forwarded from the edge camera to the fog/cloud server and then to the surveillance operation center (SOC) because it contains a foreground object. However, frames numbered 1 and 2 are discarded because they are the same as/similar

to the reference background frame labeled R or contain no foreground objects. This way, a great deal of computational and network resources saved saved.

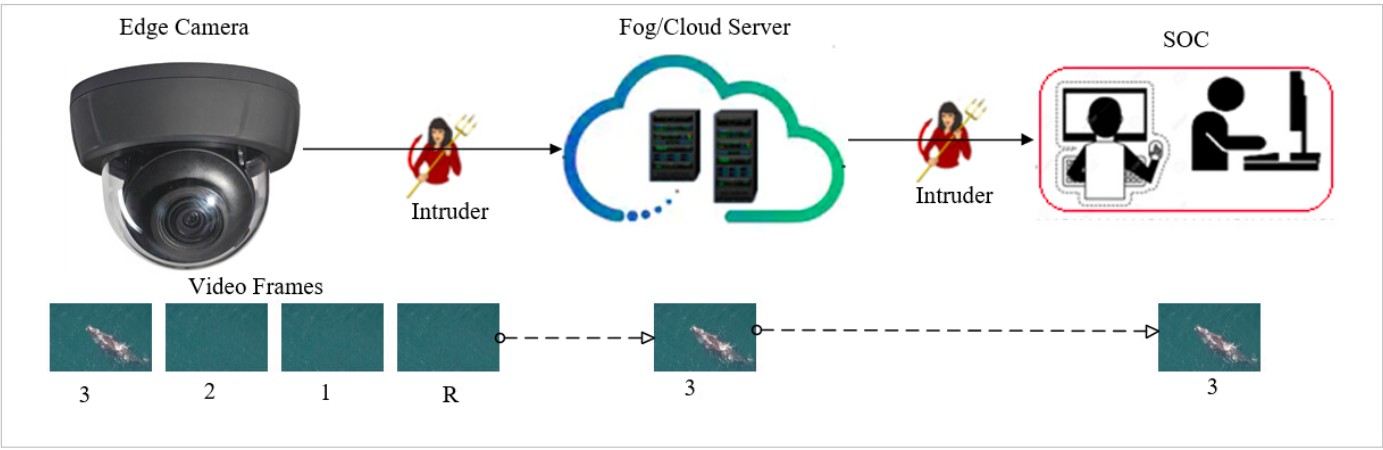

**Figure 3.** An illustration of foreground object detection and discarding of a frame containing only background object.

## 5. Reversible Privacy-Conserving Scheme

In order to at least minimize the risk of privacy invasions, individuals caught on CCTV cameras need to be anonymized. A person's face is the most powerful identifying human feature and it needs to be scrambled for the purpose of de-identification. Additionally, the people in charge of the cameras should not be allowed to unauthorizedly prowl through windows by maneuvering the cameras to observe/capture people doing private things at home. Therefore, this section mainly focuses on designing an efficient and lightweight scrambling technique for preserving the privacy of both full frames and regions of interest (face and window) on a frame. Generally, chaotic schemes are more suitable and efficient for image scrambling than other popular data encryption schemes like AES. They can be easily vectorized for pairing-pixel-wise parallel encryption process using simple but secure techniques. Hence, we developed a 2D lightweight Reversible Chaotic Masking (ReCAM) based on a dynamic chaotic system, built from a homogeneous second order differential equation of the form given in Equation (1), whose solution has unique complex roots.

$$a\frac{d^2x}{dt^2} + b\frac{dx}{dt} + cx = f(x) \tag{1}$$

Then, a chaos generator is developed following the careful investigation and analysis of the system of solutions of a constructed homogeneous second order differential equation in relation to security and computational requirements. Generally, a homogeneous second order differential equation with unique complex roots ($r_1, r_2 = \lambda \pm \mu$) is solved as portrayed in Equations (2)–(9), where $D$ is the differential operator. We arrived at the characteristic equation provided by (2) by assuming that all solutions to the differential equation will be of the form $x(t) = e^{rt}$.

$$aD^2x + bDx + cx = 0$$
$$x(aD^2 + bD + c) = 0 \tag{2}$$
$$e^{mt}(aD^2 + bD + c) = 0$$

where $x$ is represented in terms of $x_1$ *and* $x_2$ as shown in Equation (3). This is based on the basics differential equation that states that if two solutions are "nice enough" then any

solution can be written as a combination of the two solutions. In other words, $x(t) = c_1 x_1(t)$ and $c_2 x_2(t)$.

$$
\begin{aligned}
x &= c_1 x_1 + c_2 x_2 \\
&= c_1 e^{mt} + c_2 e^{mt}
\end{aligned}
\tag{3}
$$

Based on the information provided by Equation (2), the possible values of $m$ are computed by using Equation (4). The parameters $a$, $b$, and $c$ are determined through thorough research and experiments so as to ensure that a secure chaos is generated.

$$
m = \frac{b \pm \sqrt{b^2 - 4ac}}{2a}
\tag{4}
$$

Given a homogeneous equation of the form in Equation (1), there are many possible values for parameters $a$, $b$, and $c$ that can produce unique complex roots. After a thorough investigation and security analysis, we have chosen $a = 1$, $b = 4\lambda$, and $c = 4$ in this paper. Therefore, the homogeneous equation in Equation (1) becomes like the one stated in Equation (5).

$$
x'' - 4\lambda x' + 4x = f(x)
\tag{5}
$$

Now, let us deal with the unique complex roots. The foundational equation is designed such that $x = \lambda \pm i\mu$ occurs only once in the list of roots. In this case, we will only get the standard solution that contains $e^{\lambda t} cos(\mu t)$ and $e^{\lambda t} sin(\mu t)$. The values of $m$ are obtained by using Equation (6).

$$
\begin{aligned}
m &= \frac{4\lambda \pm \sqrt{16\lambda^2 - 16}}{2} \\
m &= 2\lambda \pm 2\sqrt{\lambda^2 - 1} \\
m &= 2(\lambda \pm i\mu) \\
where\ \mu &= \sqrt{1 - \lambda^2} \\
m_1,\ m_2 &= 2(\lambda + i\mu),\ 2(\lambda - i\mu)
\end{aligned}
\tag{6}
$$

The solution of the $x(t)$ in Equation (7) is finally obtained by applying the Euler's formula on the general solution of a seconder order differential equation of the form in Equation (5).

$$
x(t) = 7.4 e^{\lambda t}[A cos(\mu t) + B sin(\mu t)]
\tag{7}
$$

To produce, a 2D chaotic solution, the y-component is produced by taking first derivative of Equation (7) as ensues.

$$
\begin{aligned}
y(t) &= x'(t) \\
y(t) &= 7.4\lambda e^{\lambda t}[(A + B\mu)cos(\mu t) + (B - A\mu)sin(\mu t)] \\
y(t) &= 7.4\lambda e^{\lambda t}[C \times cos(\mu t) + D \times sin(\mu t)]
\end{aligned}
\tag{8}
$$

Then, we conducted a number of experiments on Equations (7) and (8) to determine the correct domain of each control parameters in order to build a chaotic-generator that can produce secure ciphers. At last, based on the experimental results, analytical analysis and and computational considerations, they are transformed to Equation (9).

$$
\begin{aligned}
x(t) &= ((7.4 \times 2.7^{\lambda t}[A cos(\mu t) + B sin(\mu t)])\%1) \\
y(t) &= ((7.4\lambda \times 2.7^{\lambda t}[C cos(\mu t) + D sin(\mu t)])\%1)
\end{aligned}
\tag{9}
$$

For a secure chaos to be generated with computational efficiency, the values of the control parameters in the system of equations for $x(t)$ and $y(t)$ in Equation (9) must always be within their respective domains provided in Table 1, which are determined through thorough experiments and analysis.

**Table 1.** Secure domains of the control parameter.

| Parameters | Secure Domain |
|:---:|:---:|
| $\lambda$ | $\lambda \in [0.0005, 0.0009]$ |
| $\mu$ | $\mu \in [0.999999594999918, 0.9999998749999922]$ |
| $A$ | $A \in [1, 256]$ |
| $C$ | $C \in [256, 512]$ |
| $B$ | $B = k_1 \times \frac{C - \lambda A}{\mu}$, *where* $k_1 \in [0.89, 0.999999]$ |
| $D$ | $D = k_2 \times \frac{\lambda C - \lambda^2 A - \mu^2 A}{\mu}$, *where* $k_2 \in [0.9, 0.999999]$ |

The control parameters provided in Table 1 not only affect the output chaos content but also its security; the chaos generator is not equally sensitive to all of them, though. Any change in the values of any of the eight parameters ($\lambda$, $\mu$, $A$, $B$, $C$, $D$, $k_1$, $k_2$) produces a completely different 2D chaos. However, the $\lambda$ value is the one that can have tremendous effect on the security of the chaos. The other parameters have relatively smaller impact on the security. However, all of them are used as a component of the scrambling key.

Figure 4 demonstrates that the three scatter plots, colored blue, are secure because the values of $\lambda$ and other parameters employed to generate the chaotic outputs are within the range of their respective secure domains defined in Table 1. However, the one in red color is not a good chaos; it has some patterns. This is due to the fact that the value of $\lambda$ employed is 0.04, which is out of its secure domain. Parameters $A$ and $B$ can assume any positive numeric value as big as the computer where this module runs can support, as long as the value of $\lambda$ is within the secure range. However, for computational efficiency, the values of parameters $A$ and $B$ are constrained to domains [1, 255] and [256, 512], respectively. The variable $t$ can assume a value as big as the size of the 2D-chaos intended to be generated, $t \in [0, W \times H - 1]$, where $H$ and $W$ are, respectively, the height and width of the chaos.

At last, Equations (11) show how the pixel values of the red (R), green (G), and blue (B) channels of the chaos used for scrambling a frame are generated. Each color channel uses different set of keys as portrayed in Equation (10). That is to say, a secure 2D-chaos (Equations (11) is produced for each color channel using the corresponding keys (Equation (10), $Key = [K_R, K_G, K_B]$).

$$
\begin{aligned}
K_R &= [\lambda_r, \mu_r, A_r, C_r, B_r, D_r, k_{1r}, k_{2r}] \\
K_G &= [\lambda_g, \mu_g, A_g, C_g, B_g, D_g, k_{1g}, k_{2g}] \\
K_B &= [\lambda_b, \mu_b, A_b, C_b, B_b, D_b, k_{1b}, k_{2b}]
\end{aligned}
\tag{10}
$$

$$
\begin{aligned}
R_{chaos} &= 2D\_array([Equation(9)(K_R)]) \\
G_{chaos} &= 2D\_array([Equation(9)(K_G)]) \\
B_{chaos} &= 2D\_array([Equation(9)(K_B)])
\end{aligned}
\tag{11}
$$

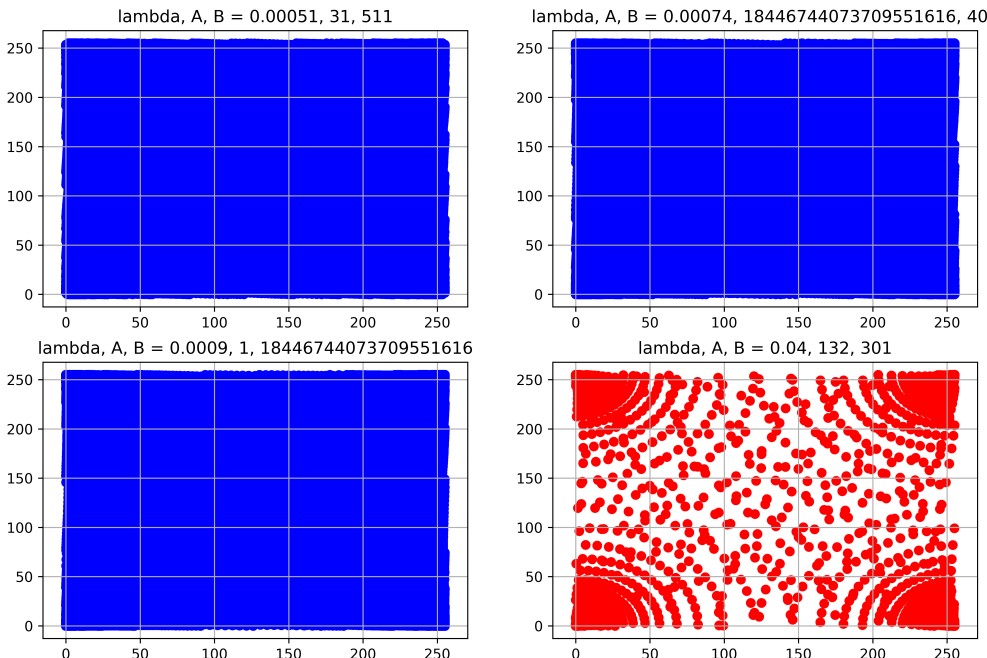

**Figure 4.** The effects of parameters' values on the security of the output chaos. The values of $\lambda$ greatly affects the output chaos in that a random enough chaos can be generated only if $\lambda \in [0.0005, 0.0009]$.

After the R, G, and B channels of an input frame have been mixed with $R_{chaos}$, $G_{chaos}$, and $B_{chaos}$, respectively, their ciphers are shuffled in a 2D block size of $32 \times 32$ using the improved version of the Fisher–Yates [53] algorithm to increase the degree of diffusion and confusion to make it next to impossible for crypto-analysts to perform any security parameter analysis on a cipher to obtain clues about its corresponding plain frame. This scrambling module is then incorporated to the simple motion-detection method portrayed in Algorithm 1 at the edge camera and the scrambling process is performed as briefly described in Algorithm 2.

---

**Algorithm 2** Frame Scrambling and its Inverse Process.

---

1: *@ Edge Camera/Sensing End*

2: *frame$_{input}$ ← vid.cam()*

3: *H, W ← frame$_{input}$.size()*

4: **procedure** GENERATEKEY

5:     *key ← secRand([λ, μ, A, C, B, D, k$_1$, k$_2$])*

6:     **return key**

7: **procedure** PRODUCECHAOS(key, H, W)

8:     *chaos ← [Equations (11)](key, H, W)*

9:     **return chaos**

10: **procedure** SHUFFLECIPHER(ch)

11:     $ch_n$ ← convert_to_dataframe(ch)

12:     ch$_n$['keys'] ← range(0, len(ch), 32)

13:     **for** i in range(0, len(ch_n), 32) **do**

14:         $j$ ← random[i, n]

15:         $ch_n[i]$, $ch_n[j]$ ← $ch_n[j]$, $ch_n[i]$

16:     **return ch_n**

17: **procedure** SCRAMBLE(chaos, f$_{input}$)

18:     $f_{input}$ ← ShuffleCipher(f$_{input}$)

19:     *frame$_{encrypted}$ ← frame$_{input}$ xor chaos*

20:     **return f$_{encrypted}$**

21: *@ Receiving End*

22: *key ← received from sending end*

23: *frame$_{encrypted}$ ← received from sending end*

24: *chaos ← produceChaos(key, H, W)*

25: **procedure** UNSCRAMBLE(chaos, frame$_{encrypted}$)

26:     *frame$_{decrypted}$ ← frame$_{encrypted}$ v.xor chaos*

27:     **return frame$_{clear}$**

---

## 6. Window and Face Detection and Denaturing

Object classification and localization helps identify sensitive parts of interest on video frames. This section focuses on DNN-based models for window and face object detection, where the main process flows are portrayed in Figure 5.

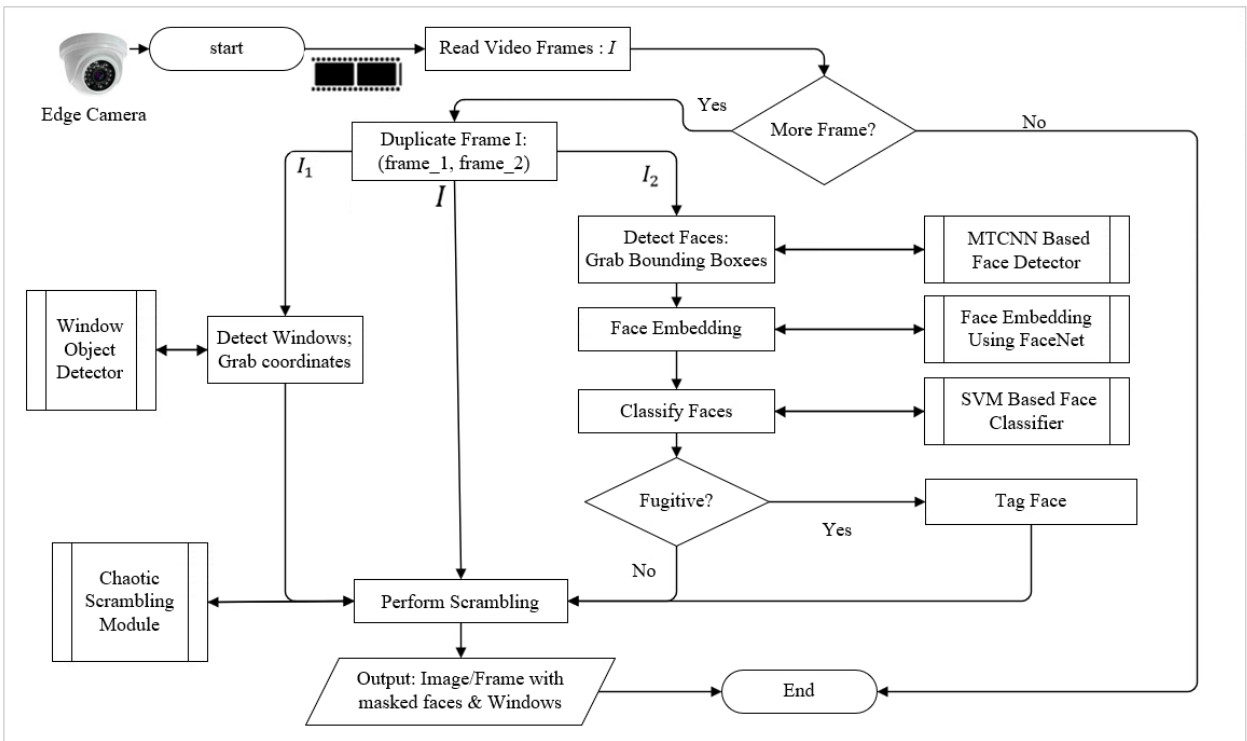

**Figure 5.** Window-and-face objects detection processes: At a server, following the decryption process, an incoming frame ($I = x(t)$) is duplicated into two more frames ($I_1$ and $I_2$). $I_1$ & $I_2$ are utilized by the window-detector & face-detector modules, respectively, to output bounding boxes if there exist(s) any window(s) or face(s). The FaceNet is used for face embeddings which are then classified by an SVM model. The chaotic module scrambles the returned windows and faces.

Window-objects consist of a number of edges that show significant changes in pixel intensity, lines (1-D structures) between image regions with same intensity on either sides, partitioned rectangularly or approximately rectangular shapes, and arcs or semicircles as portrayed in Figure 6. As a result, they could be easily detected using a combination of filtering, thresholding, edge-detection, segmentation, and contouring methods. Such an approach is lightweight and faster but it suffers from a relatively higher false positive rate.

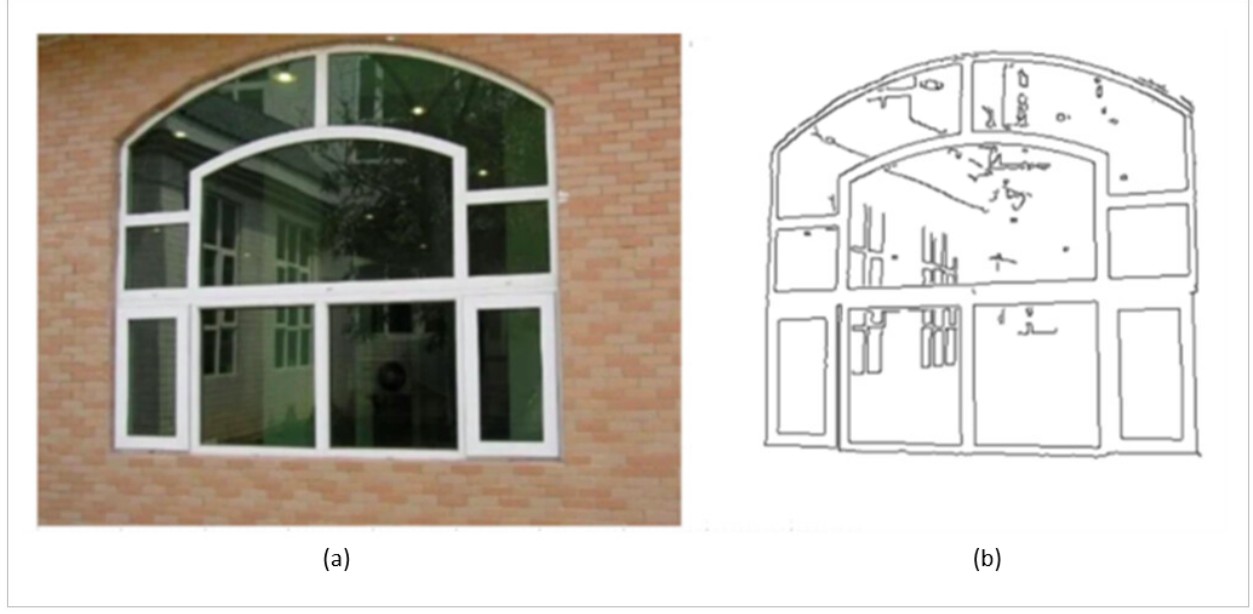

(a)                    (b)

**Figure 6.** (**a**) A sample window image, (**b**) lines, and edges that make up the window obtained after the detection process.

To train and create window-detector model on Tensorflow based on MobileNetv2 [54], we created a dataset comprising 27,000 images, out of which 50% of them contain windows and the rest are walls or images that contain no windows, downloaded and web-scrapped from the Internet because there is not a publicly available dataset for windows. Basically, sixteen types of windows are considered, which are of different types, orientations, style, color intensity, and material. An approach based on the transfer learning technique is employed which allows the generation of an accurate model using a medium image dataset size. As portrayed in Figure 7, knowledge was transferred into the target model from the MobileNetv2 network that was trained on the huge COCO17 dataset. The model construction process includes major tasks such as dataset pre-processing, training a TensorFLow model, and inference. In the pre-processing phase, the input images are first resized to a size of $320 \times 320$ so as to fit into the source model. Then, the images are labeled and mapped using the Python Label Image package. A sample of sixteen types of labeled windows is shown in Figure 8. Secondly, a model that detects window-objects is trained by leveraging the Transfer Learning technique and the TensorFlow Object Detection application programming interface (API). Once the model has been created, the inference process is achieved using the OpenCV-python. For the purpose of the transfer learning, a pre-trained model (ssd_mobileNetv2_320×320_coco17) cloned from the TensorFlow Model Garden was employed for the detector model.

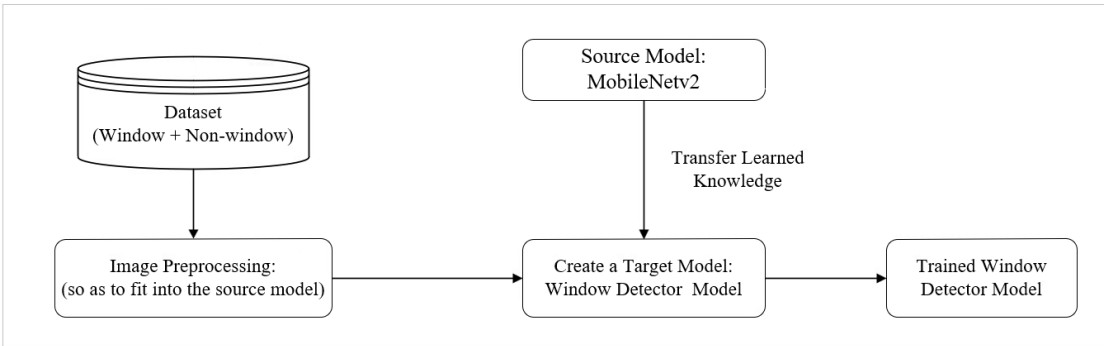

**Figure 7.** Creation a window-object detector using transfer learning.

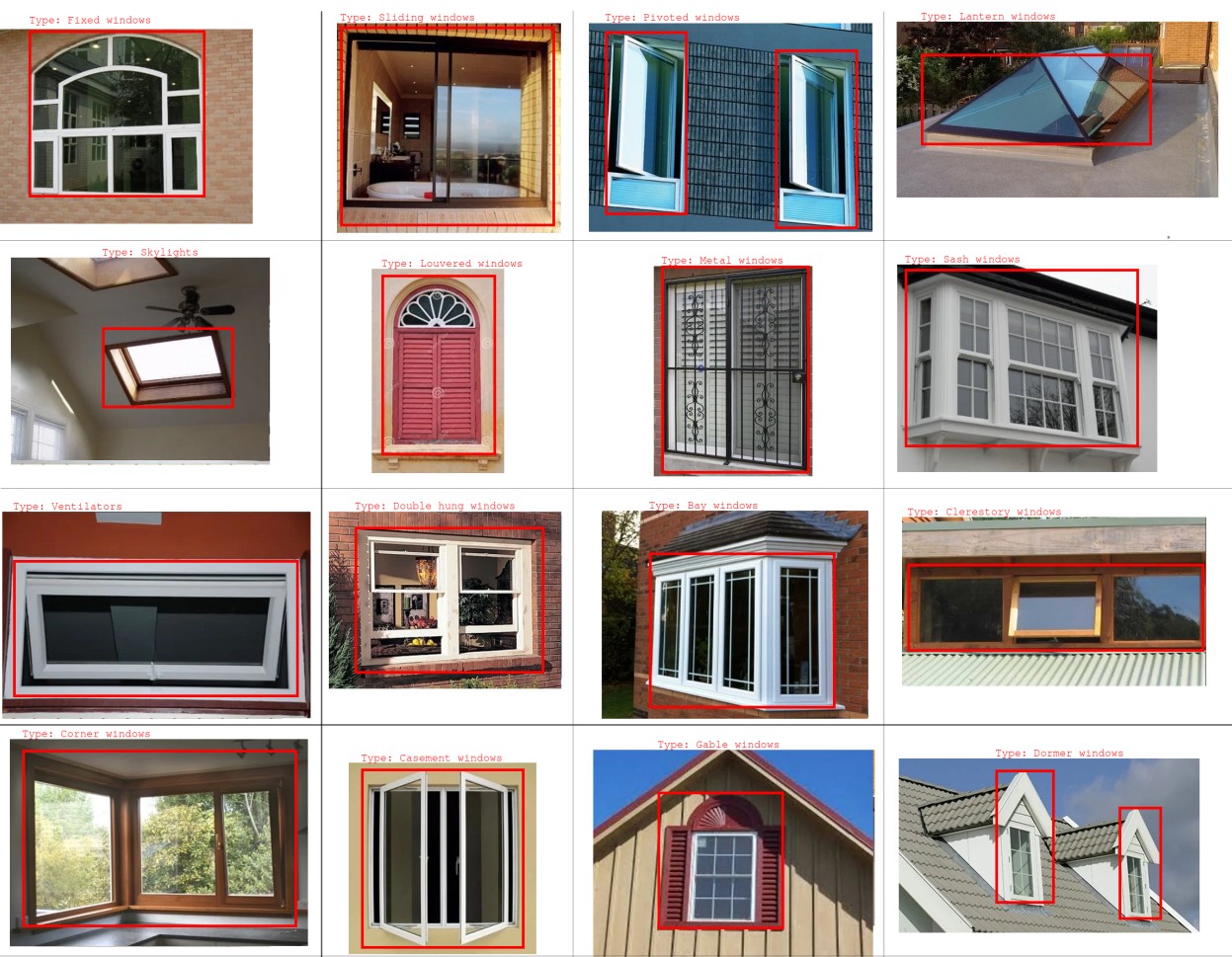

**Figure 8.** Window objects: sixteen types of windows, namely fixed windows, sliding windows, pivoted windows, lantern windows, skylights, louvered windows, metal windows, sash windows, ventilators, double hung windows, bay windows, clerestory windows, corner windows, casement windows, gable windows, and dormer windows are considered in the training and testing processes [55].

Robust object detection method is one of the top requirements of building a good privacy preserving mechanism. Hence, to automatically and accurately detect faces in video frames and localize them by drawing a bounding box around them, a robust face detector model based on the state-of-the-art MTCNN model is trained [36,37]. It can detect and extract faces in different orientations (left, up, down, and right), and even under some occlusions. As indicated in Figure 5, the face processing includes the following models:

- *MTCNN based Face Detector*: The MTCNN employs a three-stage cascaded framework. The first stage is the proposed network (P-Net) that estimates the bounding box regression vectors to calibrate the candidate faces after which non-maximum suppression (NMS) is applied to put highly overlapped faces together. The Refined Network (R-Net) is the second stage that refines false faces and carries out the bounding box regression calibration and NMS candidate merging. The output network (O-Net) is the third stage that describes the face in a more detailed manner. It outputs the five facial landmarks' positions (eyes, nose, and two mouth endings). This way, the MTCNN addresses the alignment problems in many other face detection algorithms. Then, we built MTCNN-based face-detector using TensorFlow and trained it using images from the open source Challenge of Recognizing One Million Celebrities in the Real World [56].
- *FaceNet* [35]: A deep CNN-based unified embedding for face detection and clustering that uses a triplet loss function for training. The triplets loss function is calculated

from the triplet of three pictures, Anchor (A), Negative (N), and Positive (P) images. Its main goal is to distinguish positive and negative classes by distance boundaries. It extracts high-quality 128 element vector features from the face used for future prediction and detection of faces. We employed it to create face embedding. That is, to extract the high-quality 128 element vector features from every face on a frame to be used for identifying fugitives or wanted criminals using the support vector machine (SVM) model.

- *SVM Model*: A Linear SVM model is developed to classify face embeddings as one of the wanted criminals or fugitives. The faces of wanted criminals or fugitives from justice are first detected and extracted using MTCNN followed by face embedding performed by the FaceNet model. Then, the face embedding vectors are normalized and fit into the SVM model. Hence, every incoming frame is tested using the trained SVM model to check whether it contains faces of wanted people. If a face is classified as wanted, it is tagged and is not denatured. Besides, the system alerts the authorized people about the detected wanted person.

After creating window and face objects detectors, all inferences are performed on the cloud/fog server. The window-and-face detection and denaturing processes can be performed at the edge; however, doing so consumes resources. In our PriSE design, every object-containing frame is encrypted at the edge and transmitted to the receiving end. Hence, detecting the windows or faces, denaturing them and then scrambling them along with other frame contents again during frame enciphering at the edge is unnecessarily costly. Hence, the approach pursued is that frames are transmitted from the edge cameras to the fog/cloud server in fully encrypted form. Then, the detection of window-and-face objects and subsequent denaturing are performed on the server just before they are forwarded to the viewing stations.

## 7. Key Management

Figure 2 shows that video streams are transmitted from cameras to the fog/cloud server, and then from the fog/cloud server to the viewing stations in encrypted form. Hence, a client-server architecture based lightweight agents are employed for efficiently managing the key distribution as shown in Figure 9. The key is generated in the form of list data structure, $Key = [K_R, K_G, K_B]$, where $K_R$, $K_G$ and $K_B$ the keys for the three channels of input frame. Camera, server, and viewing-station agents are developed and deployed. The camera agent stores the public key of the server agent, and generates a session key, as described in Algorithm 2, for every frame scrambling. The server agent keeps record of the public keys of all clients for the purpose of secure session key exchange. The viewing-station agent unscrambles frames to be viewed live by security personnel in a security operations center (SOC). It keeps the public key of the server for recovering a session key that is used to decipher encrypted frames forwarded by the server. In addition to the transit of video streams in encrypted form, the public keys are also employed for the exchange of security authentication to prove the authenticity and integrity of devices.

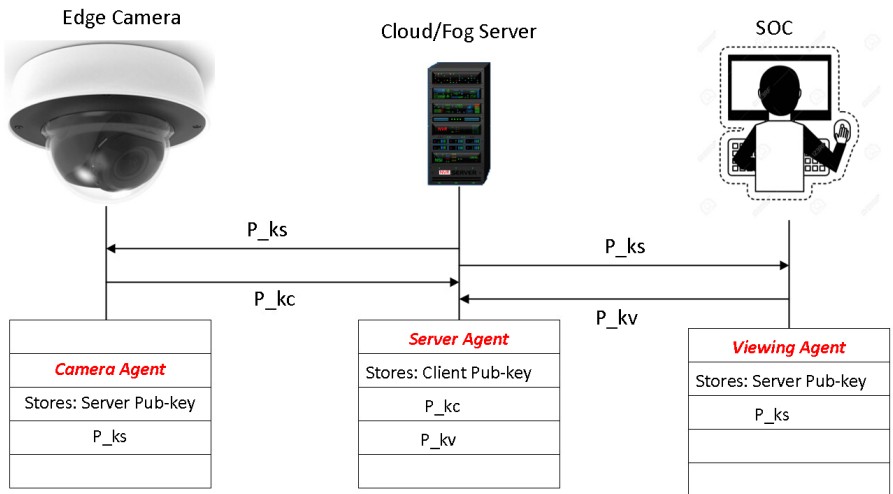

**Figure 9.** Client-server based key distribution management comprising three agents, namely camera-agent, server-agent, and viewing-agent.

## 8. Experimental Study

### 8.1. Experimental Environment Setup

Unlike the traditional cloud/fog computing based surveillance systems, the edge-enabled surveillance system migrates some computational power and intelligence to the point where the video frames are created and collected. The smart cameras at the edge are assumed to have a Raspberry Pi 4 incorporated, whose detailed specifications are provided in Table 2. These low-cost, tiny single board computers (SBC) enforce the privacy-preserving mechanisms on the video frames at the point of video frame creation to ensure end-to-end privacy protection. The implementation is done using Python 3.7.4 multithreading and multiprocessing, where the global interpreter lock (GIL) is disabled. In addition, the video frame/image used for the experiment and implementation has a size of 480P ($480 \times 640 \times 3$) and three color channels (RGB) with 8-bit wide pixels. These devices, parameters, and programming features are consistently employed throughout the experimental implementations presented in subsequent subsections.

**Table 2.** Specification of the Raspberry Pi 4.

| Parameters | Specifications |
|---|---|
| CPU type/speed | Quad core Cortex-A72 (ARM v8) 64-bit SoC @ 1.5GHz |
| RAM size | 4GB LPDDR4-2400 SDRAM |
| Integrated Wi-Fi | 2.4GHz and 5GHz |
| Ethernet speed | 1Gbps |
| Camera port | 2-lane MIPI CSI |
| Bluetooth | 5.0 |
| Power Requirement | 3A, 5V |
| Operating System | Debian Linux 10 based |

### 8.2. Simple Motion Detector Algorithm

This subsection demonstrates how the relatively static or unchanging backgrounds of video frames created by fixed CCTV cameras can be exploited to improve the processing speed,and reduce the bandwidth and storage requirements. Figures 10 and 11 demonstrate how the simple motion detection algorithm works using video frames captured by two cameras with Raspberry PI. The first frame in Figure 10a captured by PI-camera-1 is employed as a reference against which subsequent frames are compared to check if there is any motion. The next frame in Figure 10b is similar to the reference frame. As a result, their difference becomes null or a black frame illustrated in Figure 10c. However, the frame in Figure 10d contains a hand object and its difference with the reference frame is a

hand as portrayed in Figure 10e. Likewise, Figure 11 shows motion detection process on PI-camera-2. Just like the previous case, frames (a) and (b) in Figure 11 are a reference frame and a next frame similar to the reference frame respectively. As a result, their difference is null as shown in Figure 11c. However, the frame in Figure 11d has obviously a fast-moving object (blurred due to speed). Consequently, the difference between the reference frame and the frame (d) in Figure 11 is illustrated in Figure 11e.

The rule of thumb is here is that all frames whose difference with the reference frame has at least a threshold *th* of 0.29% nonzero pixels are scrambled and transmitted; otherwise, discarded. This value *th* is made smaller to make sure that no real object is skipped undetected. Less than 0.29% pixel changes in the difference frame are often attributed to environmental factors. To analyze its benefit in terms of bandwidth, storage, and processing time, we considered a video with 750 frames where objects are captured only in 135 of them. In the remaining 615 frames, only the background is captured. According to PriSE approach, the 615 frames are discarded and only the 135 are encrypted and forwarded to the cloud/fog server. This saves about 120*s* of processing delay and 184 MB of memory. Hence, an edge cameras configured with this motion detector algorithm save latency, bandwidth and storage by discarding a number of frames with no objects or with redundant information based on the threshold percentage of the difference ($frame_{diff}$) between every frame created and the reference frame computed using a fast vector operator. Besides, the PriSE scheme coupled with the scrambling module can process more than five frames per second (fps). However, if standard foreground object detectors like [51,52] are employed at the edge camera, only about 1 fps can be processed.

*8.3. Security Analysis of the Scrambling Method*

The functionality, performance, and security of the proposed video frame/image scrambling technique, ReCAM, are analyzed in detail in this subsection. The key performance indicators of any scrambling scheme are high exhaustive-key-search time, good security, and low computational time complexity. As a result, a number of parameters have been employed to comprehensively assess its performance and security. The US National Institute of Technology (NIST) randomness test suite, time complexity, key space analysis, histogram analysis, key sensitivity, Pixel sensitivity, Peak Signal to Noise Ratio (PSNR), Number of Pixels Change Rate (NPCR), Unified Average Changing Intensity (UACI), Information Entropy, and correlation analysis are considered. In addition, our proposed ReCAM scheme is compared to three other image encryption methods selected based on good speed, their being state-of-the-art, and good security. One of them is AES, the most secure and most widely used symmetrical encryption standard in today's Internet transport layer security (TLS). The other two are chaotic-based encryption schemes [42,43], abbreviated as Liu's & Tang's in subsequent uses. The Liu's scheme (with a shuffling round, $T = 2$) is a relatively lighter scheme designed based on a parameter-varied non-linear simple logistic chaotic map, which offers good speed and security unlike other higher dimension chaotic systems. The Tang's scheme is one of the most recently published chaotic encryption schemes, which is secure and designed based on low complexity of double spiral scans and a chaotic map.

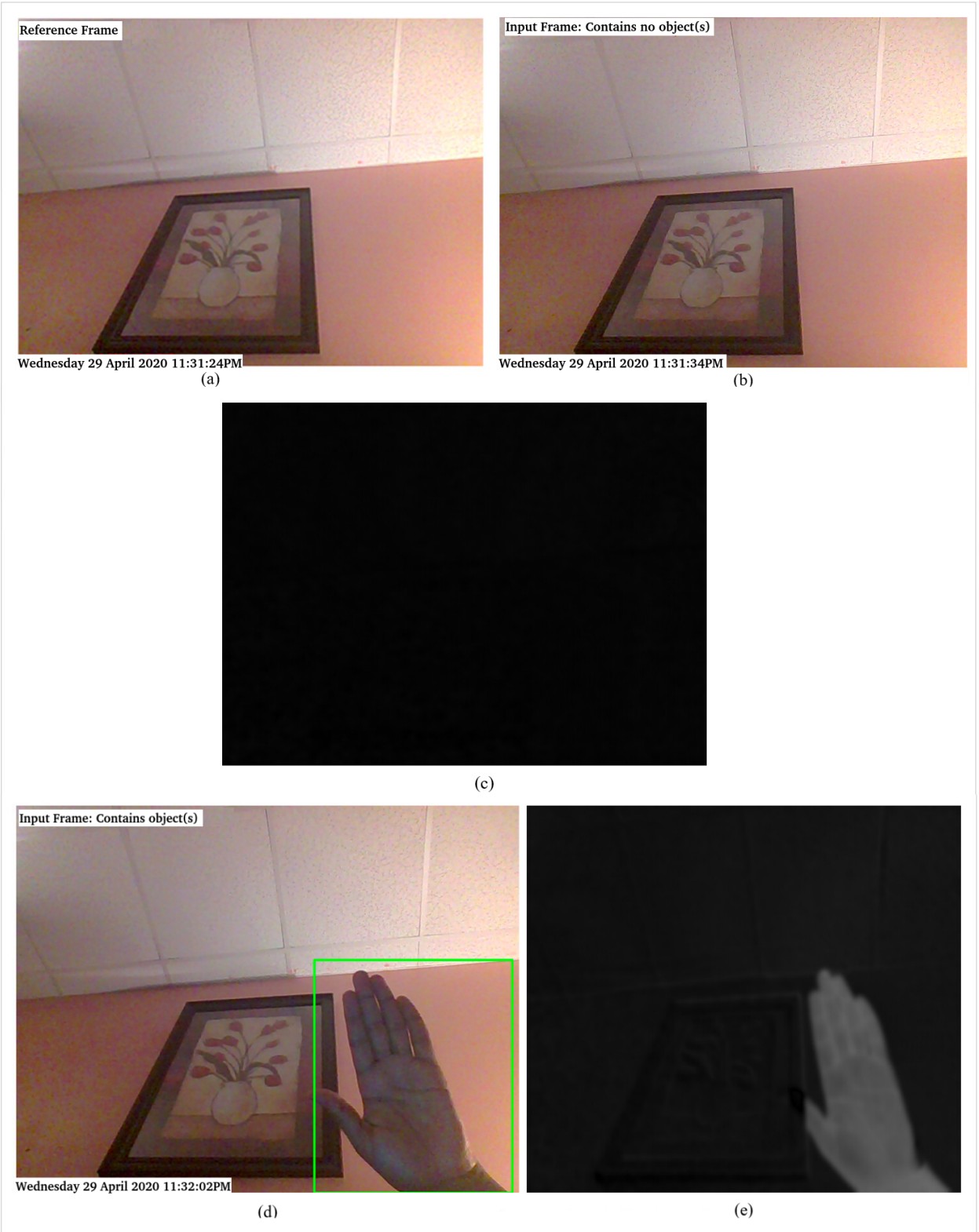

**Figure 10.** Raspberry PI-camera-1 Motion Detection: (**a**) reference frame, (**b**) next frame with no change, (**c**) a null frame which is the difference of (**a**,**b**), (**d**) frame with a hand object, and (**e**) a black-and-white frame containing only a hand object which is the difference of (**a**,**d**).

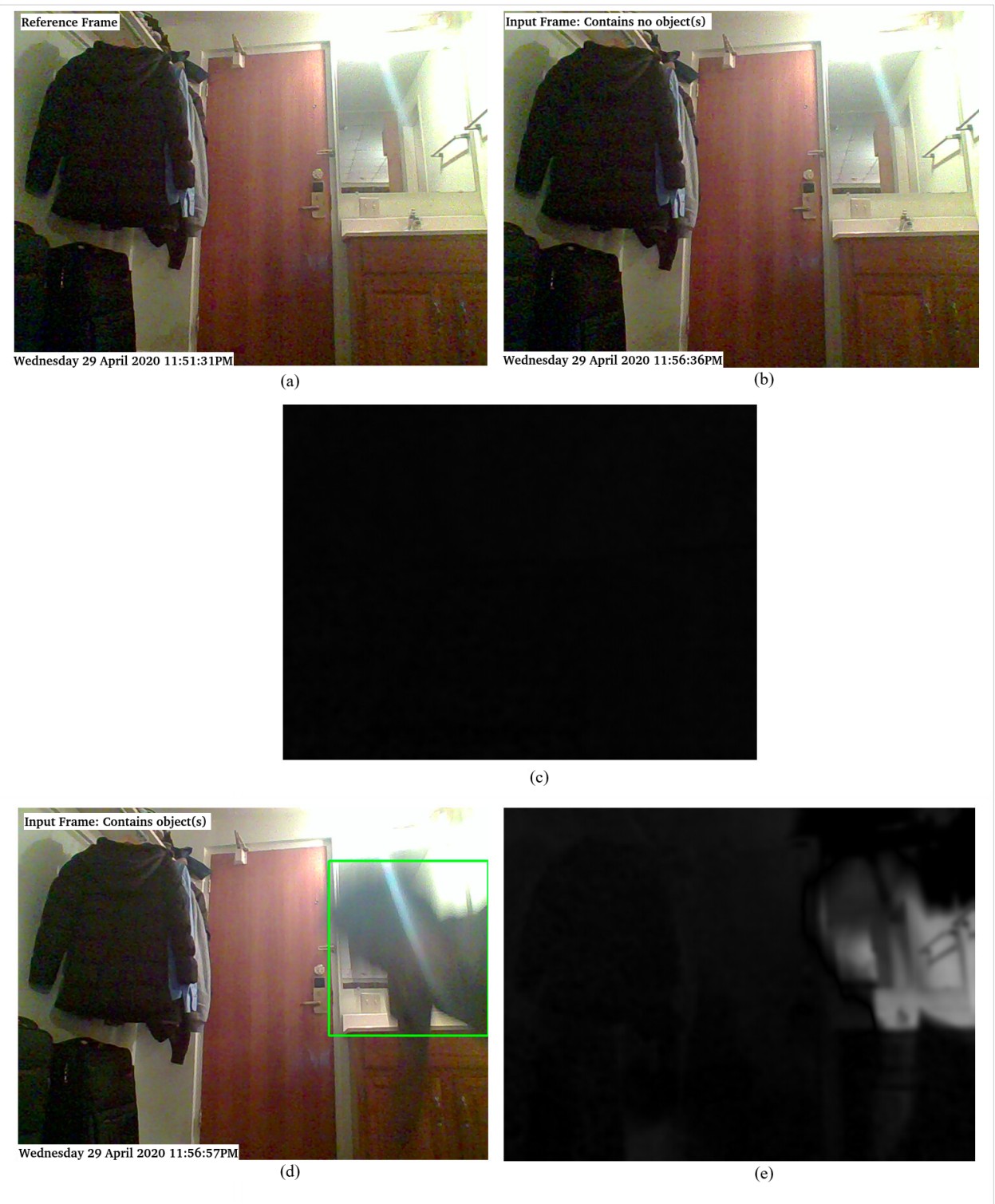

**Figure 11.** Raspberry PI-camera-2 Motion Detection: (**a**) reference frame, (**b**) next frame with no change, (**c**) a null frame which is the difference of (**a**,**b**), (**d**) frame with the shoulder and head part of a fast-moving person, and (**e**) a frame containing only the difference of (**a**,**d**).

### 8.3.1. Visual Assessment

A cipher image produced by a good scrambling scheme must not give any visual clue about its corresponding clear input image when looked at. Figure 12 illustrates an input

image and a video frame along with their corresponding ciphers. The ciphers give no visual clue about their corresponding inputs that proves the visual security of the ReCAM.

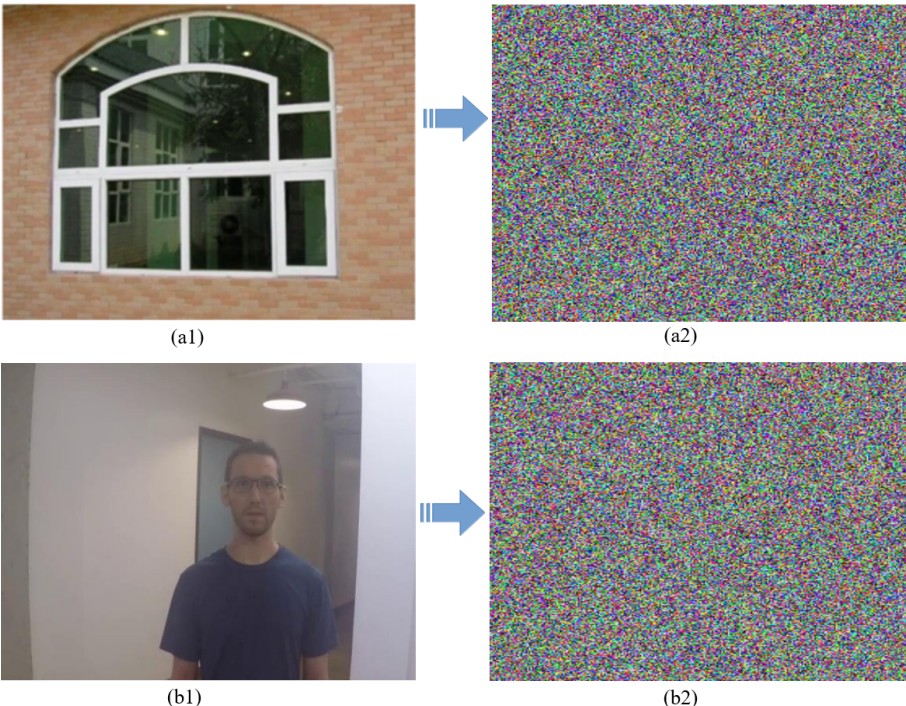

**Figure 12.** Visual Assessment: (**a1**) A plain image of a window, (**a2**) scrambled version of plain image (**a1**) which gives no clue about it, (**b1**) a clear input frame containing a person, (**b2**) encrypted version of (**b1**) which gives no information about it.

8.3.2. NIST Randomness Test Suite

The US NIST has established a test suite to evaluate the randomness of random number generators to be employed for cryptographic purposes [57]. It is a statistical package comprising 15 tests. As presented in Table 3, all the test scores are above the threshold (pass rule) 0.01 that prove the very high randomness of the ReCAM. Strong randomness implies that the encrypted frame has high degree of confusion and diffusion.

**Table 3.** Fifteen NIST randomness test results.

| Tests | *p*-Value |
|---|---|
| The Frequency (Monobit) Test | 0.532 |
| Frequency Test within a Block | 0.172 |
| The Runs Test | 0.171 |
| Tests for the Longest-Run-of-Ones in a Block | 0.842 |
| The Binary Matrix Rank Test | 0.162 |
| The Discrete Fourier Transform (Spectral) Test | 0.243 |
| The Non-overlapping Template Matching Test | 0.052 |
| The Overlapping Template Matching Test | 0.241 |
| Maurer's "Universal Statistical" Test | 0.999 |
| The Linear Complexity Test | 0.184 |
| The Serial Test | 0.502 |
| The Approximate Entropy Test | 0.602 |
| The Cumulative Sums (Cusums) Test | 0.598 |
| The Random Excursions Test | 0.667 |
| The Random Excursions Variant Test | 0.562 |

### 8.3.3. Statistical Comparison

Normally an 8-bit image comprises pixel values that range from 0 to 255 whose frequencies vary from 0 to the order of thousands depending on the image size. In an ideal situation where the pixel values of an RGB frame/image are assumed to be equally distributed, the image is supposed to have the statistics depicted in Table 4. As depicted in Table 5, the statistics of the four methods, including ReCAM, are very close to the ideal values signifying good distribution of the pixel values.

**Table 4.** Ideal statistics of an image with uniformly distributed pixels.

| Mean | STD | Min | 25% | 55% | 75% | Max |
|------|-----|-----|-----|-----|-----|-----|
| 127.5 | 73.901 | 0 | 63.75 | 127.5 | 191.25 | 255 |

**Table 5.** Comparative security and performance analysis.

| Statistics | ReCAM | Liu's | Tang's | AES |
|------------|-------|-------|--------|-----|
| Count | 921,600 | 921,600 | 921,600 | 921,600 |
| Mean | 127.499 | 127.893 | 127.743 | 127.31 |
| STD | 73.897 | 73.863 | 73.784 | 73.912 |
| Min | 0 | 0 | 0 | 0 |
| 25% | 63 | 64 | 63 | 63 |
| 50% | 127 | 128 | 127 | 127 |
| 75% | 191 | 191 | 192 | 191 |
| Max | 255 | 255 | 255 | 255 |

### 8.3.4. Histogram Analysis

Histogram analysis gives the frequency description of each unique pixel values (0 to 255) of an image. As a security requirement, the frequency of every unique pixel value is expected to be nearly equal. Figure 13a is the plain RGB-colored input image of Lenna and Figure 13b its corresponding cipher image. The histograms of the three color channels of the clear input image (a) and its cipher (b) are portrayed in Figure 13a1–a3 and Figure 13b1–b3, respectively. As can be seen from the diagrams in Figure 13a1–a3, the frequencies of the color channels of the clear image are not uniform. On the other hand, the frequencies of the color channels of the ReCAM cipher illustrated in Figure 13b1–b3 are uniformly distributed. This uniformity verifies that the scrambling scheme is robust against any statistical histogram attacks. Comparatively, the histogram of our ReCAM scheme is the best.

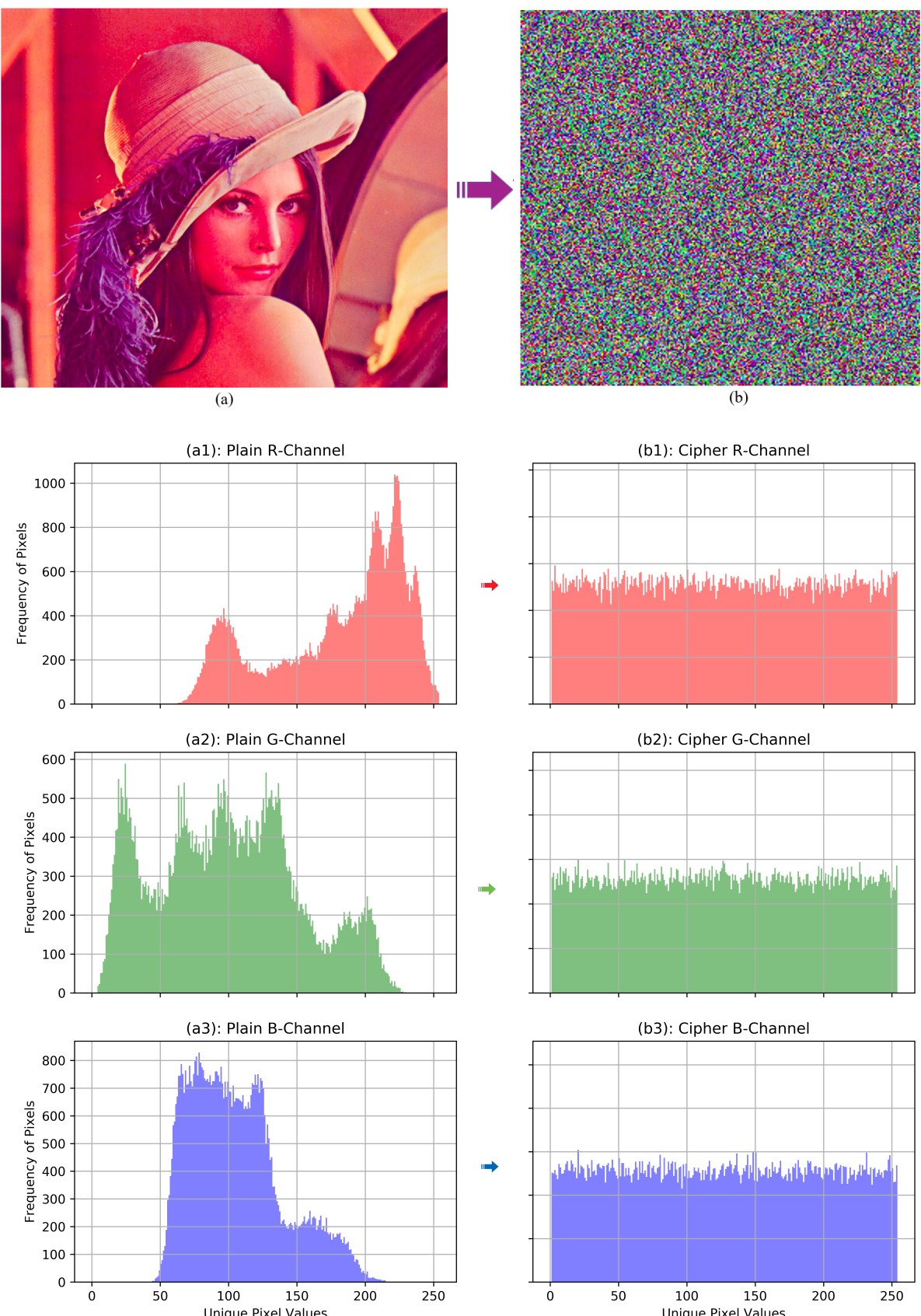

**Figure 13.** Histogram Analysis of ReCAM: (**a**) A plain input RGB-image of Lenna , (**b**) cipher of the RGB-image of Lenna (**a**), diagrams (**a1**) through (**a3**) are the histograms of the plain R, G, and B channels of clear input image (**a**), and diagrams (**b1**) through (**b3**) illustrate the histograms of the cipher R, G, and B channels of cipher image (**b**).

8.3.5. Comparative Security Analysis

On top of basic statistical values and histogram analysis, our ReCAM scheme is also compared with Liu's, Tang's, and AES techniques based on other parameters briefly described below.

- Key Space: Key space tells whether a given scheme is resistant against exhaustive-key-search analysis. The lower boundary of a secure key space of a symmetrical encryption scheme is often considered to be $2^{128}$, which is often determined by using Equation (12).

$$Key\ Space\ =\ 2^{Key\ Length\ (bits)} \tag{12}$$

- Key Senstivity: Key sensitivity measures how much the cipher changes when the key is slightly changed, in this case by only a bit. It is measured in terms of Number of Pixel Change Rate (NPCR) and Unified Average Changing Intensity (UACI). A scheme with NPCR over 99% and UACI over 33% is considered to be secure against differential analysis attacks. The UACI is defined by Equation (13), which is employed to measure the average intensity difference in a color channel between its two cipher versions $C_1(i,j)$ and $C_2(i,j)$. Besides, the mathematical definition of the NPCR, measures the change rate of the number of pixels of the cipher-frame when only a bit of the original key or pixel is modified, is provided by Equation (14).

$$UACI\ =\ \frac{1}{H \times W}[\sum_{i,j} \frac{C_1(i,j) - C_2(i,j)}{255}] \times 100\% \tag{13}$$

$$NPCR\ =\ \frac{\sum_{i,j} D(i,j)}{H \times W} \times 100\% \tag{14}$$

where $H$ and $W$ are the height and width of the cipher images, encrypted using $key_1$ and $key_2$ that vary from each other by only a bit. $D(i,j)$ is defined by Equation (15).

$$D(i,j) = \begin{cases} 1, & \text{if } C_1(i,j) \neq C_2(i,j) \\ 0, & \text{else} \end{cases} \tag{15}$$

- Peak Signal to Noise Ratio (PSNR): PSNR measures of the quality of reconstruction of a lossy compression technique, and usually the acceptable PSNR measurement falls between 33 and 50 decibels (dB). Nonetheless, it is employed in the opposite sense here. That is, the PSNR of a good scrambling scheme, where the plain image is considered as a signal and the cipher is considered as noise is expected to be lower, usually $< 20dB$. A smaller PSNR indicates huge difference between the plain image and its cipher. It is defined by using Equation (16), using the input image width ($W$) and height ($H$) and the mean square error (MSE) between pairing pixels of the plain and cipher images.

$$PSRN\ =\ 10 \times log_{10}(\frac{255^2}{MSE})$$
$$MSE\ =\ \frac{1}{W \times H} \sum_{i=0}^{W-1} \sum_{j=0}^{H-1} [I(i,j) - C(i,j)]^2 \tag{16}$$

- Entropy Analysis: The information entropy ($H(C)$), defined by Equation (17), measures the amount of randomness in the information content of the scrambled image containing $N$ pixels, where each cipher pixel is represented by $C_i$. The ideal value of

entropy for an 8-bit pixel image $I$ is $H(I) = Log_2(256) = 8$. Hence, a scheme with an entropy value very close to eight is secure against an entropy attack.

$$H(C) = -\sum_{i=0}^{N-1} P(C_i) log_2(C_i) \tag{17}$$

- Number of Frames/Images Processed per Second (FPS): The time complexity compares the efficiency of the methods over the amount of time required to process a frames per second (FPS), computed by using Equation (18). An observation period of one second is considered in this experiment. The higher the FPS, the better scheme is in terms of speed over processing steps such as encryption, detection, or classification.

$$fps = \frac{Number\ of\ frames/Images\ processed}{Observation\ period\ in\ seconds} \tag{18}$$

Table 6 shows that all the methods have comparable results in terms of key space, key sensitivity, PSNR, and entropy. If we assume that the exaFLOPS computer can perform $10^{15}$ decryptions per μs, then our scheme can be broken in $2.3 \times 10^{106}$ years by brute force analysis using this machine as illustrated in Equation (19). In general, all methods considered are secure against brute force attacks.

$$Key\ Space = 2^{448}\ bits$$
$$Time\ required = \frac{2^{448} \times 10^{-15} \times 10^{-6}}{365 \times 24 \times 3600} \tag{19}$$
$$= 2.3 \times 10^{106}\ years\ to\ break\ it$$

**Table 6.** Comparative security and performance analysis.

| Parameter | ReCAM | Liu's | Tang's | AES |
|---|---|---|---|---|
| Key Space r | $2^{448}$ | $2^{2183}$ | $\approx 2^{407}$ | $2^{256}$ |
| Key Sensitivity | | | | |
| UACI | 33.456% | 33.381% | 33.379% | 33.478% |
| NPCR | 99.673% | 99.657% | 99.642% | 99.63% |
| PSNR | 9.43 dB | 11.231 dB | 10.73 dB | 9.102 dB |
| Entropy Analysis | 7.998 bits | 7.999 bits | 7.999 bits | 7.999 bits |
| Horizontal Correlation | 0.0006 | 0.0045 | −0.00485 | 0.021 |
| Vertical Correlation | 0.0009 | 0.0039 | 0.0643 | 0.0067 |
| Diagonal Correlation | 0.0003 | 0.0054 | 0.0035 | 0.0029 |
| Speed (FPS) | 5.61 | 1.215 | 0.346 | 0.722 |

However, the ReCAM scheme stands out in terms of the cipher correlation in horizontal, vertical, and diagonal directions, and the number of frames processed every second. Our ReCAM scheme is designed to be lightweight and is much faster than the other schemes as shown in Table 6 and Figure 14. Figure 14 also shows that the FPS decreases as the frame size increases. For better processing speed and efficient bandwidth management, often smaller frame sizes like 480P and 720P are used in the practice of surveillance.

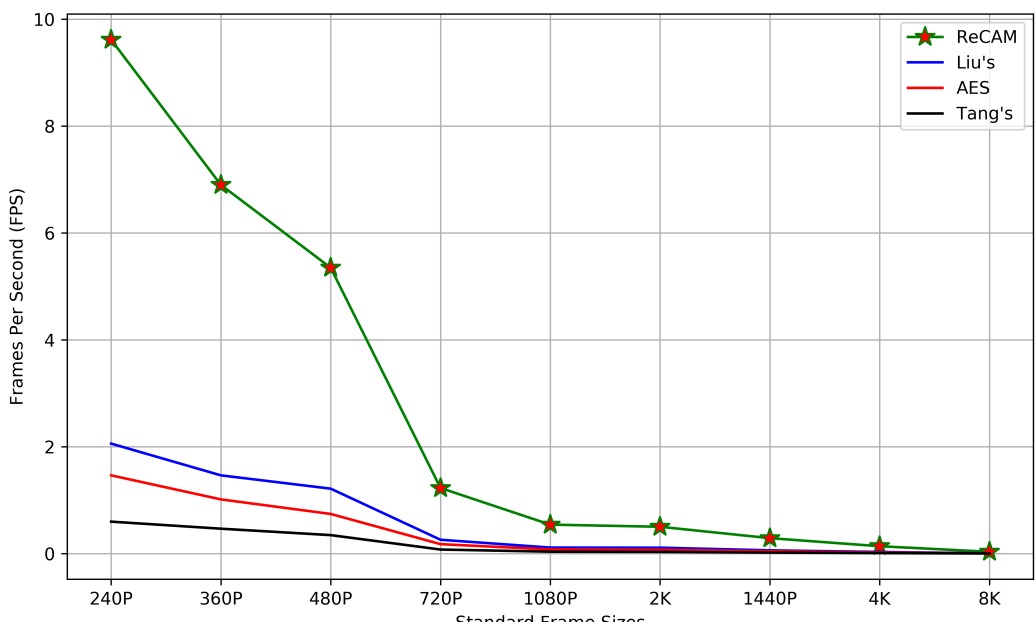

**Figure 14.** Comparison of the four schemes in terms of frames per second (fps) for different RGB frame sizes: $240P = 352 \times 240 \times 3$, $360P = 480 \times 360 \times 3$, $480P = 640 \times 480 \times 3$, $720P = 1080 \times 720 \times 3$, $1080P = 1920 \times 1080 \times 3$, $2K = 2048 \times 1080 \times 3$, $1440P = 2560 \times 1440 \times 3$, $4K = 3840 \times 2160 \times 3$, and $8K = 7680 \times 4320 \times 3$.

### 8.3.6. Overhead Analysis

In the PriSE scheme, the window and face objects detection and fugitive identification are performed on a fog/cloud server. However, the foreground object detection and whole-frame enciphering process are performed at the edge cameras where there are computational resource limitations. As a result, the edge cameras are the performance bottlenecks. The server possesses sufficient computing resources to process hundreds of frames per second. However, the edge cameras are able to process only few frames per second. Hence, it is important to optimize the bandwidth and processing time overheads introduced by the scrambling scheme. Specifically, the key generation time and the impact of the key size used for scrambling every frame is analysed in this subsection.

With regards to bandwidth utilization, the proposed ReCAM scheme has an insignificant impact because the scrambled frame is of the same size as the original plain frame. The only overhead incurred is due to the scrambling and unscrambling key, which increases the transmission size of every frame by 448 bits. The increase in bandwidth utilization due to key is computed by using Equations (20)–(23), where the input video frame size and the frame rate are employed. The frame resolution, color depth, and fps are the only parameters that affect the bandwidth utilization as far as video streaming is concerned. Equations (21)–(23) compute the bandwidth (BW) requirements of a clear frame, a cipher frame, and the key-overhead, respectively. Therefore, the result shows that the impact of the key size on the bandwidth utilization is insignificant (only 0.3135%).

$$
\begin{aligned}
Avg.\ Bit\ Rate &= Frame\ size \times Avg.\ Frame\ Rate \\
Frame\ size &= Resolution \times Color\ Depth
\end{aligned}
\tag{20}
$$

where $Resolution = H \times W$ pixels, and the $ColorDepth$ is in bits.

$$
\begin{aligned}
F_{clear}\ BW &= 640 \times 480 \times 3 \times 5.12\ \text{fps} \\
&= 4.5\ \text{Mbps}
\end{aligned}
\tag{21}
$$

$$key\_size = 7 \times 64 \text{ bits} = 448 \text{ bits}$$

$$F_{cipher} BW = (640 \times 480 \times 3 + 448) \times 5.12 \text{ fps} \tag{22}$$

$$= 4.5022 \text{ Mbps}$$

$$BW \ overhead = \frac{4.5022 - 4.5}{4.5} \tag{23}$$

$$= 0.0095\%$$

Furthermore, in terms of computational time, the key processing phase takes very small amount of time in comparison to the frame pre-processing and encryption. In other words, the key generation, session key encryption using public-key cartographic scheme for secure exchange, and key transmission process altogether take about a millisecond (ms). The key generation and session key encryption processes take about 0.3536 ms, and 0.2392 ms, respectively. The transmission time for a session key of size 448 bits is roughly computed as follows by using Equation (24). It takes about 0.5 μs.

$$Transmission \ Time = \frac{DataSize}{Camera \ Ethernet \ Speed}$$

$$Transmission \ Time = \frac{448 \text{ bits}}{\frac{10^9 \text{ bits}}{s}} \tag{24}$$

$$Transmission \ Time = 0.448 \text{ μs}$$

In summary, the proposed ReCAM scheme does not bring negative impacts to the bandwidth utilization. It rather improves the bandwidth utilization through the introduction of a simplified foreground object scanner at edge cameras, which drops frames containing only background objects. Therefore, it saves unnecessary wastage of bandwidth, processing time, and storage space. Besides, the key generation takes insignificant amount of time in comparison to the frame processing time. The scalability of the surveillance system is also improved by introducing the ReCAM scheme. It can support as many cameras as possible with smaller overheads. The performance issue is often confined to individual cameras at the edge of network. In real-world scenarios, the deployment of CCTV cameras is modular. That is, network video recorder (NVR) servers that are capable of supporting eight to 512 channels are employed. Every channel supports a single CCTV camera. Then eight to a maximum of 512 IP cameras are often connected to a single NVR server. Hence, the ReCAM scheme does not bring negative impacts on the scalability when the number of cameras is increased from eight to 512.

### 8.4. Denaturing Window and Face Objects

To prevent prowling through windows to observe and record people doing private activities in their homes by someone in charge of the cameras, all detected windows are denatured on the fog/cloud server before frames are forwarded to the surveillance operation centers. Besides, faces of individuals caught on cameras are denatured at the server for the purpose of de-identification. Figure 15 illustrates how a window- and face-containing frame is processed at the point of creation, at server, and how it is eventually forwarded to the viewing station.

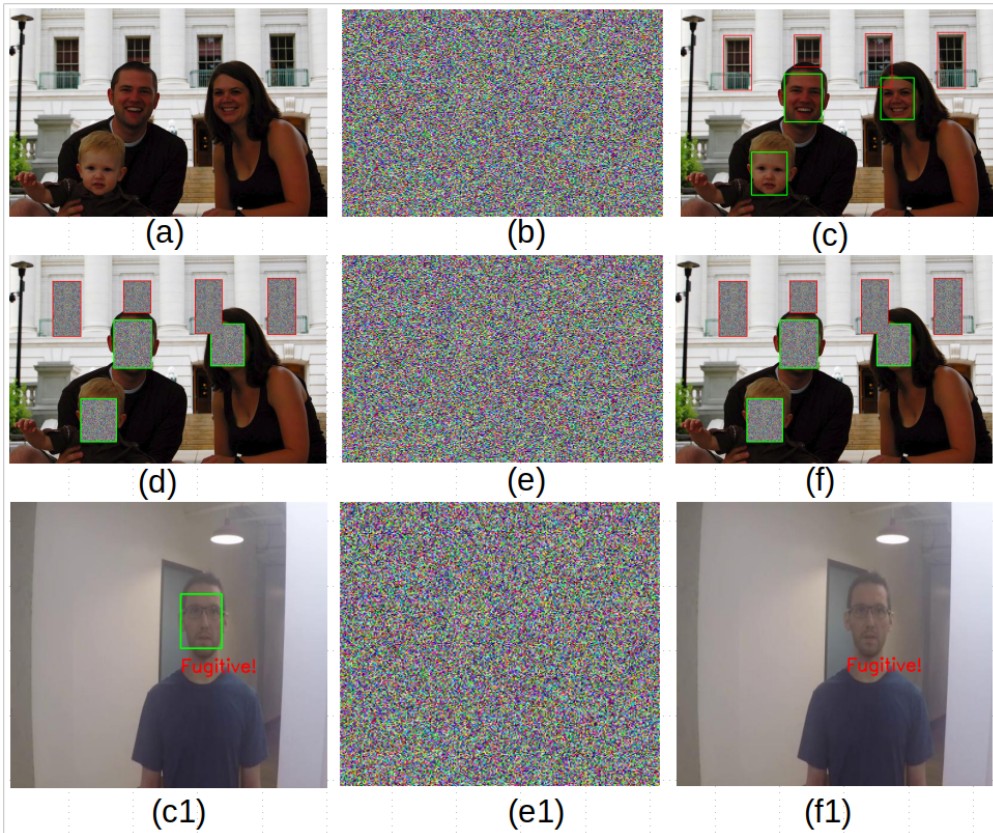

**Figure 15.** Window detection and denaturing: (**a**) an input image/frame [58], (**b**) input image/frame scrambled at the edge and in transit over a network, (**c**) the input image/frame with windows and faces detected at the server following unscrambling process, (**d**) the input image/frame with detected windows and faces denatured on fog/cloud server, (**e**) the whole image/frame encrypted at fog/cloud server and about to be sent to a viewing center, (**f**) the image/frame played to personnel at the viewing station following decryption, (**c1**) a video frame with a suspected face detected at the server (video source [59]), (**e1**) the whole frame encrypted at fog/cloud server w/o denaturing the face, and (**f1**) is the frame containing clear face (with a "Fugitive!" text mark) of suspect displayed at viewing center following decryption.

For the purpose of evaluating the robustness of the detection and classification models, four major performance metrics, namely sensitivity, specificity, accuracy, and receiver operating characteristic (ROC) curve are utilized to evaluate performance. For the analysis, the term true positive (TP) refers to the number of window objects correctly detected as windows from the given sample, false positive (FP) refers to the number of non-window objects incorrectly detected as windows, true negative (TN) means the number of non-window objects correctly identified as non-window, and false negative (FN) is the number of window objects incorrectly identified as non-window. In total, 80% of the dataset is used for training and the remaining 20% is used for testing, with equal distribution of both class labels.

- Sensitivity/True Positive Rate/Recall: The sensitivity of a window detector is its ability to correctly determine the positive class (window objects). To estimate sensitivity, Equation (25) calculates the proportion of true positive in window objects. In other words, sensitivity determines what proportion of the actual window-objects got correctly detected by the model. The window-object detector has a sensitivity measure about 98.8%, whereas the SVM based fugitive classifier has a sensitivity of 89.9%.

$$Sensitivity = \frac{TP}{TP + FN} \qquad (25)$$

- Specificity/True Negative Rate (TPR): The specificity of a window detection model is its ability to correctly determine the non-window objects. It is computed as the proportion of true negative in non-window objects by using Equation (26). Specificity deetrmines what proportion of the negative class (non-window objects) got correctly classified by the model. It could also be defined in terms of the false positive rate (FPR), which indicates what proportion of the negative class got incorrectly classified by the classifier. The specificity of the window-detector model was 99/7% and the SVM based fugitive classifier was 90.29%.

$$
\begin{aligned}
Specificity &= \frac{TN}{TN + FP} \\
&= 1 - FPR = 1 - \frac{FP}{FP + TN}
\end{aligned}
\tag{26}
$$

- Accuracy: The accuracy of a model (for example a window detector) is its ability to differentiate the window and non-window containing images or video frames correctly. To estimate the accuracy of a model, Equation (27) calculates the proportion of true positive and true negative in all evaluated cases. Hence, the accuracy measures of the window-object detector model was 99.3% and the SVM classifier was 93%.

$$
\begin{aligned}
Accuracy &= \frac{\#\ of\ correct\ detections}{Total\ \#\ of\ detections\ made} \\
&= \frac{TP + TN}{TP + TN + FP + FN}
\end{aligned}
\tag{27}
$$

- Receiver Operating Characteristic (ROC) Curve: The ROC curve plots the TPR (sensitivity) against FPR (1-specificity) at various threshold values and essentially separates the 'signal' from the 'noise'. The Area Under the Curve (AUC) is the measure of the ability of a detector to distinguish between classes and is used as a summary of the ROC curve. The higher the AUC, the better the performance of the model at distinguishing between the positive and negative classes.
  As portrayed in Figure 16, the window-object detector and the SVM classifier have higher areas under their respective curves that signify good performance at detecting windows and identifying fugitive faces, respectively.

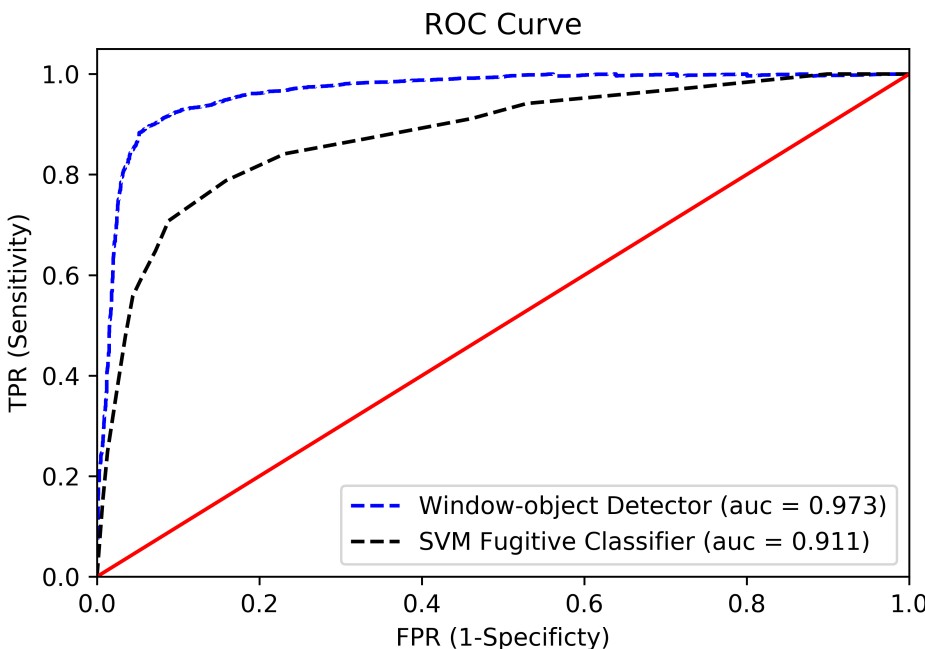

**Figure 16.** ROC curves for the window-object detector fugitive classifier.

Summing up the evaluation, the custom-made window object detector model built for the PrISE is able to successfully detect more than 99.3% of the various types of windows tested due to their easy nature. The face-detector is highly robust that can detect faces with an average accuracy of 99.5%. However, the SVM-based face classifier has an accuracy of 93%, a bit smaller than that of the face-detector.

### 8.5. Overall Performance of the PriSE Scheme

The entire PriSE system (face/window selection, fugitive identification, scrambling, communication) is analyzed for system level performance of timeliness. From Table 6 (bottom row), the PriSE scheme enables more fps and hence, an efficient privacy-preserving video method at the edge. The overall processing speed at the edge of the proposed system is 5.12 fps, when coupled with the simplified motion detection scheme, scrambling, and cloud/fog/edge communications.

The whole frame enciphering and window-and-face objects detection could be done at the edge camera. However, there is limited computing resources at the edge; as a result, detecting the windows and faces, denaturing them, and encrypting the whole frame at the edge is not economical. The better way to do it is to perform the whole frame scrambling at the edge following object motion detection, and to perform the window-and-face objects detection and denaturing at the fog/cloud server. No content of the frame, including the windows and faces, is disclosed over the communication channels. As shown in Figure 15, windows and faces are detected at the server, denatured, and whole frame is re-encrypted and forwarded to the viewing station. At the viewing station, the whole frame is decrypted but the windows and faces remain denatured to prevent peeping and ensure anonymity. However, faces of fugitives, as identified by the SVM model, are not denatured. Rather they are forwarded to the SOC as they are as shown in Figure 15f1 for immediate actions. Generally, it is possible to reverse all encrypted faces and windows if needed. The scrambled windows and faces can be unscrambled on arrival at the SOC through the exchange of their respective keys. However, to reverse scrambled windows and faces on stored videos, an additional requirement for secure storage of encryption keys must be met. The latter is out of the scope of this paper. Nonetheless, for the secure storage and sharing of such keys and efficient management of distributed access to stored

surveillance videos, we are currently researching to leverage the blockchain technology in order to introduce permissioned blockchain-based solutions.

Today's publicly available object-detection methods can only process less than 3 fps at the edge with computing power similar to that possessed by Raspberry Pi devices. Besides, the best encryption schemes we have today can only encrypt less than one fps at the edge. Our proposed ReCAM scrambling scheme is able to process 5.61 fps on the edge device using Raspberry PI 4, which is better than the preexisting schemes but it still has limitations. However, we are witnessing the quick development of more powerful single board computers (SBC) and it is reasonable to expect a satisfactory performance on next generation edge devices.

## 9. Conclusions

According to our research so far which includes discussions with law enforcement, the privacy breaches by the preexisting practice of video surveillance are mainly attributed to four factors, namely (1) interception attacks, (2) abuse of CCTV cameras by people in charge, (3) indiscriminate practice, and (4) the problem of balance between privacy and usability.

In this paper, we have proposed a novel slender video frame scrambling scheme to ensure end-to-end privacy of frame contents and built a window-and-face object detection to identify and denature windows and faces on frames to prevent the abuse of CCTV cameras for voyeurism or peeping through windows and to anonymize individuals caught on camera. The privacy-preserving surveillance as an edge service (PriSE) was demonstrated with superior performance compared to available video security methods. The security and computational efficiency of the Reversible Chaotic Masking (ReCAM) scrambling technique was tested and analyzed extensively using standard cryptographic analysis techniques and apropos performance metrics. Besides, the window-object detection method was trained and tested using various types of windows. The model loaded on fog/cloud server can detect windows with an accuracy of 99.3% and the MTCNN based face-detector can detect faces with an accuracy of 99.5%. In addition, our pragmatic experimental results show that the scrambling scheme can process 5.61 fps on a Raspberry PI 4. However, when coupled with the simplified motion detection scheme, the overall processing speed at the edge becomes 5.12 fps. Hence, the extensive experimental studies and analyses carried out validate that the PriSE is able to efficiently scramble frames and recognize and denature windows to prevent unauthorized image content in real-time.

**Author Contributions:** Conceptualization, A.F. and Y.C.; Data creation, A.F. and Y.C.; Formal analysis, A.F., Y.C. and S.Z.; Investigation, A.F., S.Z., E.B. and G.C.; Methodology, A.F., Y.C., E.B. and G.C.; Software, A.F., Y.C. and S.Z.; Supervision, Y.C., E.B. and G.C.; Validation, A.F., Y.C. and E.B.; Writing—original draft, A.F., Y.C. and S.Z.; Writing—review & editing, E.B. and G.C. All authors have read and agreed to the published version of the manuscript.

**Funding:** This research received no external funding.

**Conflicts of Interest:** The authors declare no conflict of interest. The views and conclusions contained herein are those of the authors and should not be interpreted as necessarily representing the official policies or endorsements, either expressed or implied, of their institutions.

## Abbreviations

The following abbreviations are used in this manuscript:

| | |
|------|------|
| ACLU | American Civil Liberties Union |
| AES | Advanced Data Encryption |
| AUC | Area Under Curve |
| CAN | Campus Area Network |
| CCTV | Closed Circuit Television |

| CPU | Central Processing Unit |
| DNN | Deep Neural Network |
| E2E | End-to-End |
| FPR | False Positive Rate |
| FPS | Frames Per Second |
| GIL | Global Interpreter Lock |
| GSM | Global System for Mobile Communications |
| HOG | Histogram Oriented Gradient |
| LFW | Labeled Faces in the Wild |
| ML | Machine Learning |
| MTCNN | Multi-Tasked cascaded Convolutional Neural Network |
| NIST | National Institute of Technology |
| NMS | Non-Maximum Suppression |
| NPCR | Number of Pixels Change Rate |
| NVR | Network Video Recorder |
| PSNR | Peak Signal to Noise Ratio |
| PriSE | Privacy-preserving Surveillance as an Edge service |
| PTZ | Pan-Tilt-Zoom camera |
| ReCAM | Reversible Chaotic Masking |
| ROC | Receiver Operating Characteristics |
| SBC | Single Board Computers |
| SMS | Short Message Service |
| SOC | Surveillance Operation Center |
| SVM | Support Vector Machine |
| TLS | Transport Layer Security |
| UACI | Unified Averaged Changing Intensity |
| WAN | Wide Area Network |

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
