# Peer review of "Privacy-Preserving Surveillance as an Edge Service Based on Lightweight Video Protection Schemes Using Face De-Identification and Window Masking"

_electronics, doi:10.3390/electronics10030236_

Round 1

Reviewer 1 Report

Authors provide the research on a really important topic of privacy insurance in CCTV video surveillance, that could be of great interest to the readers. Still I see some major points that should corrected before accepting the manuscript for publications. Please find below my remarks grouped into minor (more technical corrections) and major (principal, that should be improved). Major: 1. Provide more details on the dataset of windows generated (size, does it contain non-window objects, proportion of window/non-window objects, normalization methods applied, etc.); please also provide the downloadable link to the dataset for independent evaluation. 2. Provide more technical details on "8.3. Security Analysis of the Scrambling Method", when it comes to the tests applied. 3. When describing "8.4. Denaturing Window and Face Objects" no description of the experiments performed (dataset preparation, model architecture and parameters, FNRs/FPRs, ROC, training time, etc.) are given except the final accuracy. Please expand. 4. The same comment (as 3) is relevant for describing the use of SVM for criminal face detection. 5. Discuss the question of reversibility, i.e. if it is possible to restore the initial frames if such need arises. Minor: 1. Improve the quality of pictures (Fig. 1, 3, 5) 2. Apart from Fig. 2, which is very illustrative, please also provide the general method and architecture diagrams, using standard UML notations. 3. Redraw Fig. 5 using standard UML notations.

Reviewer 2 Report

The paper is interesting, technically sound and well-written. The idea and contribution are original and relevant.

My only concern is about the idea of "privacy". Why did you decide it was enough to protect faces and windows to keep privacy levels? Have you reviewed the relation between your proposal and the relevant regulations such as European GDPR?

I suggest the authors to consider these questions before the acceptance.

Round 2

Reviewer 1 Report

Authors have addressed the comments from the previous review round. Manuscript can be accepted. I think there are several small problems related to the picture quality, that can be corrected during the manuscript preparation for publishing phase.